



# The Earth's climate lagged, recurrent and non-linear solar and lunar multi-millennial scale responses: An oceanic hypothesis, evidence, verifications and forecasts

Jorge Sánchez-Sesma[1]

[1]Independent researcher, Cuernavaca, Morelos, 62440, México

*Correspondence to*: Jorge Sánchez-Sesma (jorgesanchezsesma@yahoo.com)

**Abstract.** This work provides a hypothesis of the links between the multi-millennia scale recurrent solar and tidal influences and Earth´s climate lagged responses, associated with the oceanic transport mechanisms with a variable modulation. As a part of this hypothesis, empirical and simple, non-linear lagged models are proposed for five of the most representative
Earth´s climate variables (a continental tropical temperature, an Antarctic temperature [at James Ross Island], the Greenland temperature, the global temperature and the southeast asian monsoon) with multi-millennia records to account for the lagged responses to solar forcing. The proposed models implicitly include a well-known oceanic heat transport mechanism: the Ocean Conveyor Belt. This oceanic mechanism appears to generate a climate modulation through the intensity of the ocean/atmosphere circulation, and a heat and mass transport, with a consequent climate lag of several thousands of years.
Tidal forcing is also considered for global temperature modelling and forecast. The consequent millennia-scale global forecasts, after being integrated/verified with an accumulated ocean travelled distance from the tropical East Pacific, and with a double evaluation of the tidal influences based on similarities and on the NASA's solar system astronomical dynamics, indicates a cooling for the next century, and gentle oscillations over the next millennia. Our preliminary results that strongly suggest that millennial scale changes in solar activity induce circulation and thermal global impacts, also
suggest that the Younger Dryas event, may be influenced by the lagged outcomes of solar driven changes in the tropical Pacific, and by tidal influences. The detected Earth´s climate delayed responses, that have been working in the past and present climates, and will be working in the future climates, must be, as soon as possible, independently verified and theoretically sustained, before to be fully included in a multi-scale climate models as a scientific theory. A final example for the global temperature record over the last 170 years demonstrates with experimental results for the twenty first century
evolution the convenience of a multi-scale climate modelling with contrasting lower values compared with the IPCC global temperature scenarios.

## 1 Introduction

  *"In a short preliminary review of the scientific work of the Norwegian-British-Swedish expedition, E.F. Roots points out that the lag between climatic change and change in form of the glaciers may be longer than the period of climatic change*
*itself."*

***Glacier Variations and Climatic Fluctuations (Ahlman, 1953).***

*"The Sun is a yellow dwarf star, a hot ball of glowing gases at the heart of our solar system. Its gravity holds the solar system together, keeping everything – from the biggest planets to the smallest particles of debris – in its orbit. The connection and interactions between the Sun and Earth drive the seasons, ocean currents, weather, climate, radiation belts*
*and auroras.*

***Our Sun (https://solarsystem.nasa.gov/solar-system/sun, 2021).***

  *"Tidal forcing is very weak and an unlikely candidate for millennial variability; the Keeling and Whorf proposal is considered as the most likely among unlikely candidates."*

***Millennial Climate Variability: Is There a Tidal Connection?(Munk et al., 2002).***



During the last decades the global community has been considering the anthropogenic climate change (IPCC 2001, 2007, 2013) as one of the most serious threats for humanity. However, in the same decades several reconstructions of astronomical and Earth´s climate (EC) processes, have provided enough information to review and update several important aspects of solar and climate variabilities. For instance: recurrent solar forcing (SF), glacial and interglacial climatic processes, and their
lagged responses (LRs) associated to oceanic dynamical and thermal processes, have been carefully reconstructed over the last millennia.

Another important aspect is that all the SF processes present are associated with LRs, that have been documented thanks to the reconstructions of solar and EC processes developed during the last decades. For instance, a) Sánchez-Sesma, (2016; hereafter SS16) detected a recurrent and potentially lagged response of solar activity to the planetary gravitational forces
based on cosmogenic radionuclide-based reconstructions of solar activity and NASA's solar dynamics simulation over millennia; this lag was estimated to be greater than 6 Kyr; b) Shackleton et al. (2000) and Shakun et al. (2015) [hereafter S00 and S15, respectively], have detected LRs of the climate system to orbital and SF. In particular, S15 have analyzed the last 800 Kyr of ocean climate, and estimated important lags for the sea water isotopic anomaly ($\delta^{18}O_{sw}$) record, respect to the sea-surface temperature (SST) variability. For the 23, 41 and 100 Kyr orbital oscillations associated with precession, obliquity
and eccentricity, S15 estimated their lags of ~3, 5 and 20 Kyr, respectively.

In this work, thanks to those mentioned reconstructed SF and EC records, the solar influences on EC are analyzed. To do this analysis, the LRs are accepted as an intrinsic part of the EC system that are strongly linked to climate processes. Thus, the EC LRs are part of complex processes that can be assesed taking into account recent reconstructions of SF and EC events over different scales covering the last millennia. This work proposes the surface part of the well known "ocean conveyor
belt" (OCB)" as the main mechanism of the non-linear LR of the EC. The results obtained emphasize the existence and importance of various processes associated with the recurrent multi-millennial SF and its lagged responses through the OCB. The results also suggest an important contribution of the SF LRs to the variability of the following climate processes: a) a tropical continental temperature from the Congo River Basin (crbT), b) the Northeast Antarctic Peninsula temperature from the James Ross island (jriT), c) the Greenland temperature (GrT), d) the Global Temperature (GT) and e) the south asia
monsoon index (samI), with increasing lags to solar activity of ~0.09, 2.2, 3.9, 4.05, 4.45 Kyr, respectively. These increasing lags, associated with increasing oceanic accumulated distances from the tropical East Pacific, are suggesting the persistent great influences of the oceanic transport processes. These lagged processes, after being verified with a simple distance-lag model, support extrapolations of our findings to future times.

A large fraction of the EC variability can be explained by the solar and lunar lagged forcings allowing for a multi-millennial
experimental forecast. For instance, global temperature in the 21st century is experimentally explained with multi-millennial solar and lunar variability that suggest a cooling for the next centuries; however this explanation is improved when the multi-decadal variability (also suggested to be a lagged solar influence) is included, suggesting a cooling for the next decades.

This past and future multi-scale modeling framework suggest that further research is needed to better understand the LRs as a part of a complex oceanic-dependent climate processes in different temporal scales.

## 2 Background

Several important aspects about climate are presented here as background. Some of these aspects are about the long-term orbital forcing, and their lagged responses, and others are related to climate modelling.

### 2.1 Orbital and millennia scale forcing

Milankovitch (1940) calculated the astronomical variations of seasonal solar radiation supply and pointed out that, with the
disposition of large landmasses and oceans as in the last million years, the difference of summer radiation at about 65°N might determine the occurrences of ice ages and warm interglacials. The same author characterizes three main orbital forcings: eccentricity, obliquity and precession, with periodicities around 100, 41 and 23-19 Kyr, respectively. However, there are recent and complementary efforts to reconstruct shorter periodicities, the multi-millennia scale, of the total solar irradiance (TSI) based on isolated and combined isotopic sources, tree-rings and ice-cores (Solanki et al., 2004; Steinhilber
et al. 2009; Steinhilber et al. 2012). After analyzing these and other records, from 9 to 40 Kyr length, cosmogenic





radionuclide-based reconstructions of solar activity with a linear modeling of their oscillations, Sánchez-Sesma (2016; hereafter SS16) detected a 9.5 Kyr recurrent influences, with a consequent multi-millennial solar forecast. This forecast, that includes a grand-minima for the next centuries, is in agreement with several alternative forecast methods (vgr., Steinhilber & Beer, 2013), and is supported by astronomical information that suggests a possible planetary gravitational forcing of solar
activity, by an unknown mechanism (SS16); (See Figure 1).
Keeling and Whorf (hereafter KW; 1997, 2000) also suggested that the millennial scale climate variability is related to extreme oceanic tides. They argue that these extreme tides that are associated with orbital coincidences reoccurring at certain repeated periods, generate cooling of the sea surface by increased vertical mixing. KW2000 following a hypothesis put forward by Pettersson (1930), detect recurrent tidal patterns of around 1800 and 4680 years. In particular, the 4680 year
pattern is associated by KW2000 with the temporal distance between the Sun-Earth-Moon syzygy and Earth´s perihelion occurrences (See Figure 2).

*2.2 Ocean circulation and abrupt climate changes*

An application of the $^{14}$C method is the determination of the rate and detection of movement of deep ocean water. Preliminary results, published by Kulp (1953), from samples taken on cruises by the Woods Hole *Atlantis* and the USN
*Hydro San Pablo* in the summer of 1951, indicated deep-water age of about 1600 to 1700 years at two sites in the North Atlantic (55°19'N; 32°57'W and 53°53'N; 21°06'W), and pointed out: "if this sample waters started near the surface of the Arctic, also indicate that the rate of of turnover of the oceans must be on a scale of at least 10,000 years long. An exact knowledge of this important oceanographic parameter should certainly be of interest in the study of world-wide climatic changes. Pulses of particularly old centuries migth be reflected in the climate some thousands of years later as the water
returns to the surface."
The deep ocean water circulation, that is part of the Ocean conveyor belt (OCB) [a concept developed by Broecker (1991)] has had an important science-fiction introduction: *Twenty Thousand Leagues Under the Seas: A World Tour Underwater* (*Vingt mille lieues sous les mers: Tour du monde sous-marin*). It is a classic science-fiction adventure novel by French writer Jules Verne, published in 1870, that explores deep seas in the world. With this kind of literature works, Verne and other
writers, have promoted experimental ocean campaigns in the scientific world. For instance, efforts coming from the Britain's Challenger expedition, a massive four-year undertaking begun in 1872, that vastly increased knowledge about the deep sea, had a goal: to circumnavigate the globe and study the ocean's depths. It is important to mention that 20,000 leagues are around 111,120 kms (See Figure 3).
In 1991 Wallace Broecker wrote: "A diagram depicting the ocean's "conveyor belt" has been widely adopted as a logo for
the Global Change Research Initiative (GCRI). This diagram (Fig. 3b)....was designed as a cartoon to help the largely lay readership of this magazine to comprehend one of the elements of the deep sea's circulation system..... The lure of this logo is that it symbolizes the importance of linkages between realms of the Earth's climate system. The ocean's conveyor appears to be driven by the salt left behind as the result of water-vapor transport through the atmosphere from the Atlantic to the Pacific basin. A byproduct of its operation is the heat that maintains the anomolously warm winter air temperatures enjoyed
by northern Europe...... A millennium of very cold conditions known as the Younger Dryas (YD), appears to have been the result of a temporary shutdown of the conveyor. Thus the conveyor logo portrays the concern that led to the launching of the GCRI: that complex interconnections among the elements of our Earth's climate system will greatly complicate our task of predicting the consequences of global pollution."
There are several abrupt climate change events. Geological evidence, mainly from Greenland, reveals natural occurrences of
large, abrupt climatic changes that are not uncommon (Alley, 2000). For instance, the YD, a rapid return to near-glacial conditions lasting nearly a millennium during the last deglaciation, is an example of abrupt climate changes.

*2.3 Long-term climate reconstructions and lagged responses*

Emiliani (1966) analyzed with radio-isotope methods the last 420 Kyr of the Caribbean waters providing enough objective information about climate variability of SST. Lamb, Lewis and Wodroffe (1967) based on Milankovic and Emiliani works,
compared insolation versus the SST and pointed out: "The ocean temperature curve appears to respond not to strictly





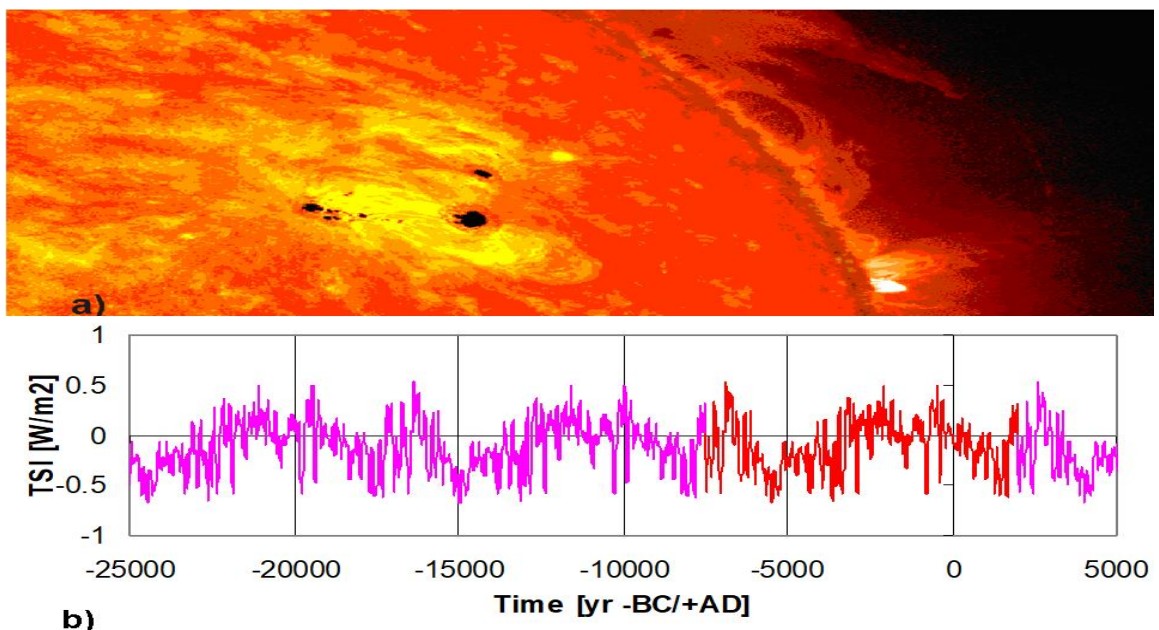

**Figure 1:** Graphical display of solar activity and its estimated recurrent processes. a) An active region in the sun with dark sunspots. [Image Source:NASA/SDO/ AIA/HMI/Goddard Space Flight Center], and b) Total solar irradiance (TSI) over the last 27Kyr and the next 3 Kyr, accepting the detected recurrence of around 9.5Kyr (Sánchez-Sesma, 2016; SS16). Original integrated record in red color, and extrapolated (forward and backward) in cyan color.

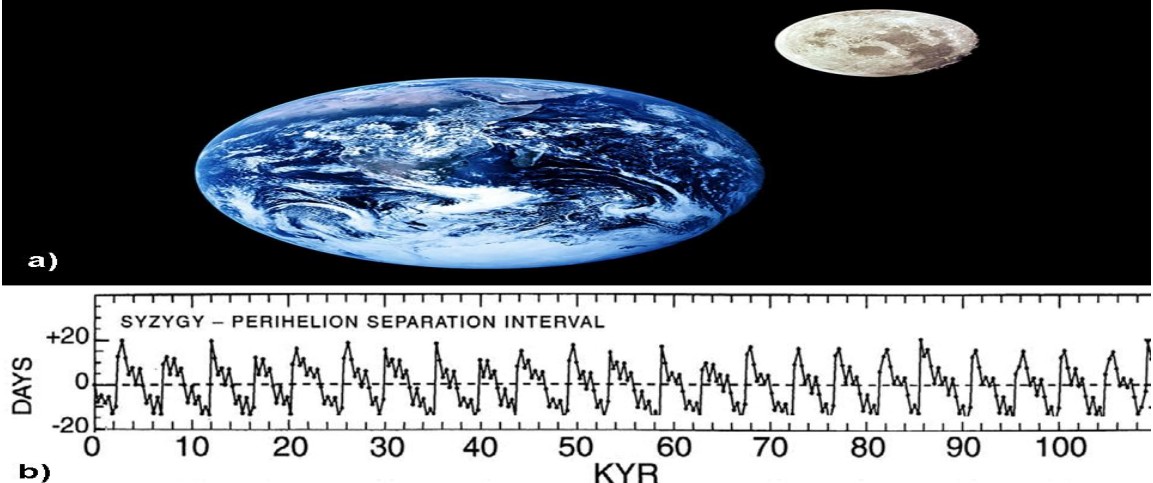

**Figure 2:** Graphical display of the Earth and moon, and their estimated tidal recurrent processes. a) An artistic image showing the Earth and the Moon, and b) Temporal diference between syzygy, (an alignment of the Earth, Sun and Moon) and perihelion (point in the orbit of the Earth when it is nearest to the sun) that generates tidal oscillations of around 4700 years, Figure Adapted from KW2000 (PNAS).



contemporary radiation conditions but to the features of the 65°N radiation curve some 5,000-6,000 years earlier—a lag that might be imposed by the time taken to heat and cool the ocean water to all depths and all latitudes [cc=0.65; p=0.0025].” The same authors, after considering sea level reconstructions over the Holocene, also pointed out: “..suggest that the rising world sea level lagged somewhat behind the temperature changes...”

*2.4 Limitations of climate modelling*

It is important to mention that the current climate approach, is limited, because between several limitations, it does not consider the climatic millennia scale climatic contributions. Therefore, the current climate models have consequently presented limitations, for instance, normal global climate variability happenning in the middle of Holocene, has been reconstructed with different sets of proxy records (Marcott et al., 2013) and modelled considering all the forcings (Liu et al.,
2014). The problem arises when these records do not match both in values and trends, as Liu and coworkers, have pointed out: “A recent temperature reconstruction of global annual temperature shows Early Holocene warmth followed by a cooling trend through the Middle to Late Holocene. This global cooling is puzzling because it is opposite from the expected and simulated global warming trend due to the retreating ice sheets and rising atmospheric greenhouse gases. Our critical reexamination of this contradiction... points to potentially significant biases in both the seasonality of the proxy
reconstruction and the climate sensitivity of current climate models.”

*2.5 Multi-scale climate modelling (MSCM)*

Considering that climate oscillations cover a wide range of temporal scales: the daily (0–25 h), the monthly (25 h–0.5 year), the annual (0.5–2.5 year), the interannual, (2.5–10 year), the decadal/secular (10–400 year), the millennial (400–10,000 year), the orbital [or Milankovitch] (10,000–1,000,000 year) and the tectonic (1–600 million year) scales, that are
characterized by soli-lunar tidal oscillations, solar oscillations, terrestrial orbital oscillations, and galactic oscillations linked to the journey of the solar system around the galaxy (Scafetta, 2021), a multi-scale approach is strongly suggested. In particular, Lohmann et al. (2020) have pointed out: “The Earth's climate is characterized by many modes of variability in the atmosphere, ocean, cryosphere, and biosphere.” The same authors mentioned that: Detection of low-frequency variability in the climate system is crucial to allow a separation of low frequency from high frequency signals, thereby increasing the
ability to recognize and improve the attribution of climate events to climate forcing (external or internal).
The processes of the multi-scale climate modelling (MSCM) have been initiated with the climatic orbital influences modelling proposed by Milankovitch around 1930. This first contribution, associated with orbital influences, has provided elements to consider the relatively low-frequency climate oscillations (with periods going from $10^4$ to $10^6$ years) with recurrent and lagged responses. This orbital contribution to the MSCM, has provided information and models for important
low-frequency processes that generates the glacial-interglacial climate oscillations, fueled mainly by planetary orbital influences on Earth. Here the glacial parameters associated with ice-sheets and polar glaciers have been considered to better climate modelling.
A second contribution, has been provided by the “traditional” climate approach developed since 1950s that includes mainly annual, multi-annual and decadal climate variabilities, with its relatively high-frequency oscillations (with periods going
from 10 to $10^2$ years).
However, there are important intermediate-scale contributions to climate, that should be considered a third contribution, associated with millenia-scale oscillations, as those provided by solar variability and the ocean conveyor belt (OCB) paradigm (Broecker, 1991), which have not been considered fully in the modelling of climate and oceanic processes. Later in this work they will be empirically considered and modelled.
The sum of these three contributions to climate variability, in a rigorous approach, will require a non-linear interaction of their contributions, however, in a practical approach the usual unidirectional influences, or energy flow, from lower to higher frequencies, will facilitate their modelling as a simple sum of these contributions.
In order to detect the EC LRs to SF, we analyzed five of the most representative solar, tidal, and EC reconstructed records coming from recent studies. See Figures 1, 2 and 4.


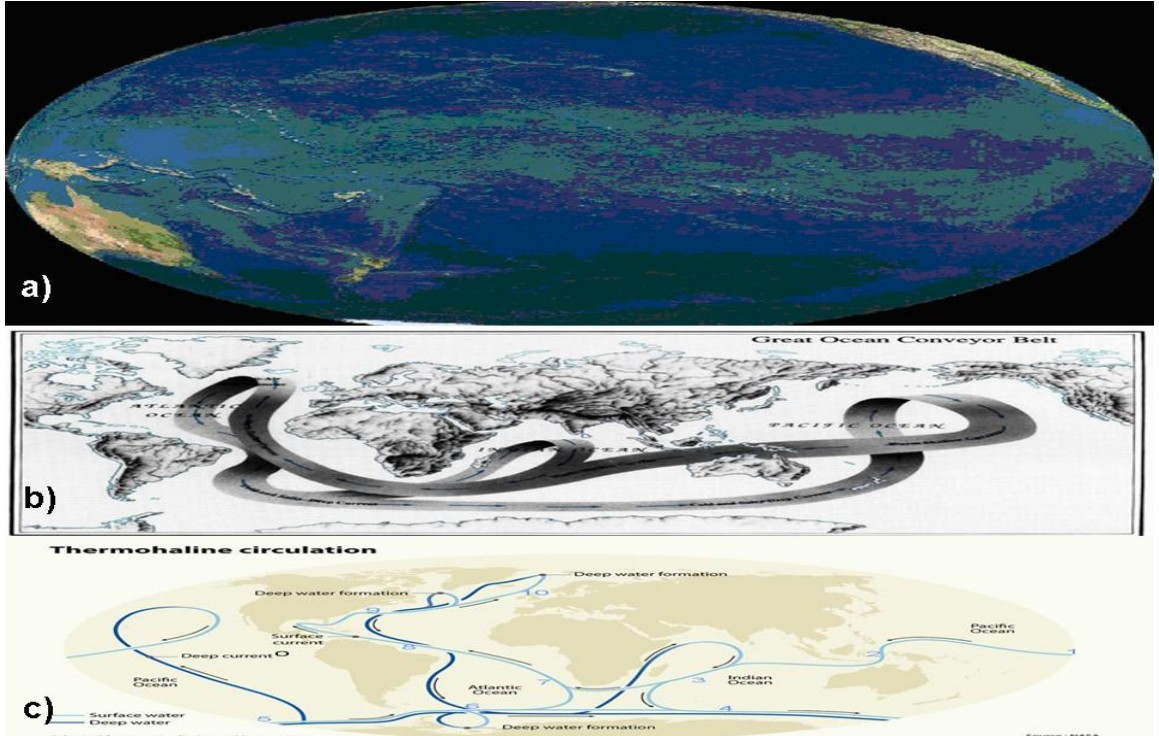

**Figure 3: Graphical display of the oceanic realm and its main circulations. a) A geostationary satellite equatorial view from central Pacific; b) an artistic view of the ocean conveyor belt (Broecker, 1991); and c) the ocean conveyor belt in a more recent representation by Laura Margueritte (NASA), where surface (deep) currents are indicated with light blue (dark blue) color. Accumulated surface travelled distances from Eastern Pacific are also displayed based in a unit of around [EqDiam/4=10,000kms].**

## 3 Materials and methods

### 3.1 Data

*Solar Forcing (SF).* Based on three solar reconstructed records of total solar irradiance (TSI) established by Steinhilber et al. (2009), Steinhilber et al. (2012), Solanki et al. (2004), that after to be integrated and analyzed in SS16, is depicted in Figure 1b. The recurrence of about 9.5 Kyr is, in time, backward and foreward extrapolated, in this graphic over the last 27 and future 3 Kyr, respectively.

*Tidal Forcing (TF).* Records of TF established by Keeling and Whorf (2000), are depicted in Figure 2b. The approximated recurrence of about 5 Kyr is shown over 30 Kyr.

It is important to mention that, in this work it is also indirectly included, as an internal forcing, the stronger and persistent oceanic characteristic, the transport of mass and heat, that is well demonstrated in the main movement of the oceans, the OCB, depicted in Figure 3.

*Earth's Climate (EC).* Here is presented the reconstructed records of five important terrestrial phenomena (See Figure 4):

    a)    A proxy reconstructed record for the continental tropical temperature of the Congo river basin (Tcrb, record 1) over the last 22 Kyr, reconstructed by Weijer et al. (2007), see Figure 4a. This information is coming from Eastern Atlantic deep





sea core sediments just in front the Congo River discharge, and is obtained through lipids deposited in soil and transported by the river to deep sea.

b) A climate reconstructed, for the northeast Antarctic peninsula, NEAP, at the James Ross Island (JRI), temperature (jriT, record 2) record over the last 14 Kyr, reconstructed by Mulvaney et al. (2012) See Figure 4b.

5     c) A proxy record of Greenland temperature (GrT, record 3) from the ice-cores covering the last 50 Kyr, that was reconstructed by the GISP2 project (Alley et al., 2000), see Figure 4c.

d) A multi-proxy reconstructed global temperature (GT, record 4) record over the last 22 Kyr, reconstructed by Marcott et al. (2013; herafter M13) (for the Holocene), and Shakun et al. (2012; hereafter S12) (for the deglaciation) see Figure 4d. This GT record (S12, M13) is adjusted taking into account both the northern and southern hemisphere temperatures 10     (NHT and SHT), with instrumental (HadCRUT5) and proxy reconstructions developed by Jones and Mann, 2004, Moberg et al., 2005 and Neukom et al., 2014 (See details in **Appendix A**).

e) A proxy record of southern asian monsoon index (samI, record 5) from the isotopic information from calcite covering the last 10 Kyr, that was reconstructed by Wang et al. (2005), see Figure 4e.

*3.2 Modelling*

15     Firstly, a model for the lagged and modulated linear contribution of a solar proxy variable is proposed as follows:

$$C_I(t) = M_{IS}[\alpha_{IS} S(t - \delta_{IS}) + \beta_{IS}(t - t_1) + \gamma_{IS}] + e_{IS}(t), \tag{1}$$

Where $C_I(t)$ is the climate variable $I$, $S(t)$ is the solar proxy variable, $M_{IS}$ is a variable linear modulation factors, $\alpha_{IS}$ is the amplification factor, $\delta_{IS}$ is the lag, $\beta_{IS}$ is the slope, $\gamma_{IS}$ is the additive constant, $t_1$ is the initial times for the modeled period, and $e_{IS}(t)$ is the error of this model.

20     Considering that the lagged responses of the EC events to solar activity are linked to the ocean transport processes, their accumulated oceanic distance traveled, D can be modelled as follows:

$$D = a_3 \delta^3 + a_2 \delta^2 + a_1 \delta^1 + a_0, \tag{2}$$

Where, $D$ is the accumulated oceanic distance from eastern equatorial Pacific, that oceanic fluid has travelled along the OCB, $\delta$ is the correspondent lag of the climatic variable respect solar activity, $a_n$ is the polynomial factor, and its 25     order/exponent, $n$.

The self-similar model for the tidal influences on climate is defined as:

$$C(t) = M_{SS}[\alpha_{SS} C(ta) + \beta_{SS}(t - t_1) + \phi_{SS} + e_S(t)], \tag{3a}$$

with     $t > t_{r1}$; and     $ta = \dfrac{(t - t_1 - \delta)}{\gamma} + t_1$,     $(ta-t1)G = t -d -t1$     (3b)

and where, $M_{SS}$ is a variable modulation factor, $\alpha$ is the amplification factor, $\gamma$ is the time scale factor, and $t_1$ and $t$ are the 30     initial times for the analysis and the modeled periods, respectively.
The *Mss* is proposed as a variable precession oscillation; it is defined as:

$$M_{SS} = \alpha \sin(2\pi(t-\delta)/21000) + \beta(t-t_1) + \gamma \tag{4}$$

Where $\alpha$ is the amplification factor, $\delta$ is the phase lag, $\beta$ is the slope, $\gamma_{IS}$ is the additive constant,

There is an objective form to evaluate the tidal effect on Eath. It is the solar and lunar aceleration vector, $g(t)$ [m/s$^2$], is given 35     by (Bueche, 1975):





$$\vec{g}(t) = [\text{GM}_S / \left\| \vec{r}_{ES}(t) \right\|^2] \, \hat{r}_{ES}(t) + [\text{GM}_M / \left\| \vec{r}_{EM}(t) \right\|^2] \, \hat{r}_{EM}(t) \tag{5}$$

where $GM_S$ and $GM_M$ are the gravitational constant multiplied by the solar and lunar mass, with values of $1.3271244004 \times 10^{20}$ and $4.902800066 \times 10^{12}$ [m³/s²], respectively.

$\vec{r}_{EA}(t)$ [$\hat{r}_{EA}(t)$] is a vector [unit vector] expressed in [m] directed from the Earth to the astronomical mass A (Sun[S] or Moon[M]). The positive force is attractive and points toward the attractive sources (Sun or Moon).

In all these models, parameters are estimated through iterative or multi-linear regression processes that minimize the RMS values of errors. Instead of developing statistical analysis as convergence and confidence level estimations, we prefer in this stage of research on climate recurrences, to apply verification/replication of all of our findings with independent information in our estimation processes and results. It takes into account both the dating limitations and the approximated values provided by all proxy reconstructions,

Future climate reconstructions with more accurate information will provide better elements for further and refined statistical analysis.

## 4 Results

Here we present results for the five selected EC variables. At first, the continental climate represented by the Tcrb record is modelled by a solar forcing with a variable modulation (Eq. 1) given by M=1 for 0<t<10KyrBP, M=1+0.133(t-10) for 10<t<17.5KyrBP, and M=2 for 17.5<t<25KyrBP). The model (Based on Eq. 1, with, $\alpha_{1S}$=1.5 [°C.W/m2], $\delta_{1S}$ =80 [yrs], $\beta_{1S}$ =-0.00001 [°C/yr], $\gamma_{1S}$ = 0.3 [°C] ) and its extrapolations are displayed in Fig 5a (See details of the lagged and recurrent solar influences in **Appendix B**).

For second instance, the TSI record is also employed to explain and forecast the jriT record. The solar modulation employed to explain the crbT record is also employed for the *jriT* modelling. The model (Based on Eq. 1, with, $\alpha_{2S}$=2.0 [°C.W/m2], $\delta_{2S}$ =2300 [yrs], $\beta_{2S}$ =-0.00008 [°C/yr], $\gamma_{2S}$ = -0.1 [°C] ) and its extrapolations are displayed in Fig 5b (See details of the lagged and recurrent solar influences in **Appendix B**).

For third instance, the TSI record is also employed to explain and forecast the GrT record. The solar modulation employed to explain the crbT record is also employed for the *GrT* modelling. The model (Based on Eq. 1, with, $\alpha_{3S}$=12.0 [°C.W/m2], $\delta_{3S}$ =3850 [yrs], $\beta_{3S}$ =-0.0 [°C/yr], $\gamma_{3S}$ =2.0 [°C] ) and its extrapolations are displayed in Fig 5c (See details of the lagged and recurrent solar influences in **Appendix B**).

For fourth instance, the TSI record is also employed to explain and forecast the GT record. The solar modulation employed to explain the crbT record is also employed for the *GT* modelling. The model (Based on Eq. 1, with, $\alpha_{4S}$=0.5 [°C.W/m2], $\delta_{4S}$ =4070 [yrs], $\beta_{4S}$ =-0.0 [°C/yr], $\gamma_{4S}$ =0.0 [°C] ) and its extrapolations are displayed in Fig 5d (See details of the lagged and recurrent solar influences in **Appendix B**).

For fifth instance, the TSI record is also employed to explain and forecast the samI record. The solar modulation employed to explain the crbT record is also employed for the *GT* modelling. The model (Based on Eq. 1, with, $\alpha_{5S}$=-1.0 [°C.W/m2], $\delta_{5S}$ =4450 [yrs], $\beta_{5S}$ =0.0002[°C/yr], $\gamma_{5S}$=-7.6 [°C] ) and its extrapolations are displayed in Fig 5e (See details of the lagged and recurrent solar influences in **Appendix B**).

The modelling of the LRs for the mentioned five climate variables with recurrent solar forcing, generates a consequent five forecasts for the next millennia. Details of these modelling and forecasts are displayed for the past and next millennia in Figure 6.

The lagged responses to SF detected of around 80, 2300, 3850, 4070, 4450 yrs for the crbT, jriT, GrT, GT and samI, respectively, have been integrated in a lag-accumulated/distance model, depicted in Fig. 7a. Using the Eq. 2, the input pairs of data, that are displayed in Table 1, are modelled using a polynomial model and the consequent very slow velocities, are shown in Table 2.



**Figure 4: Climate reconstructed records. a) Tropical continental temperature, Tcrb for the last 22 Kyr (Weijer et al., 2007); b) Northeast Antarctic Peninsula temperature, jriT (Mulvaney et al., 2012); c) Greenland temperatures, GrT (Alley et al., 2000); d) Global Temperature, GT (90S-90N), for the Holocene and Deglacial periods (Marcott et al., 2013; Shakun et al., 2012); and e) the south asia monsoon Index, samI, (Wang et al., 2004).**



Figure 5: The climate records (see Figure 4) and their modelling and forecasts. The models are based o Eq. 2 and on the TSI records (SS16) that are displayed in red color.



**Figure 6: Zoom of Figure 5 with five modelling and forecasts over the last and future 2000 years.. The modells are based on the TSI records (SS16), are displayed in red color.**







**Figure 7:** Ocean conveyor belt accumulated distance from Eastern tropical Pacific (ETP) and its climatic lags. a) model AccOCB-Dist and lags; b) an empirical verification of the OCB proceesses originated in the ETP going around Antarctic and Arctic, based on NHT and SHT reconstructed by Shakun et al., (2012). The SHT record was lagged 1400 years and linearly adjusted to the NHT record.



**Table 1: Estimated pairs of data (Accumulated distance-Lag) for the models shown in Figures 5a, 5b, 5c, 5d and 5e. With underlines and italics are indicated the two supposed pairs of values, initial and final, considering symmetry in the subsurface and deep, OCB travel, and the multi-millennial solar cycle (SS16) forcing.**

| Region/Variable | Time Lag[Ky] | AccDist[EP/4] |
|---|---|---|
| *EEpacT* | *0* | *0* |
| jriT | 2.3 | 4.5 |
| GrT | 3.85 | 10 |
| GT | 4.07 | 10 |
| saml | 4.45 | 11 |
| *EEpacT* | *9.5* | *21* |

**Table 2: Polynomial Model Accumulated distance-Lag (AccDist-Lag) from data shown in Figure 7a**

| Time Lag | AccDist[EP/4] | V [Km/yr] | V [m/hr] | V [mm/s] |
|---|---|---|---|---|
| 0 | 0.00 | 11.39 | 1.30 | 0.36 |
| 1 | 1.60 | 20.35 | 2.32 | 0.65 |
| 2 | 3.98 | 26.80 | 3.06 | 0.85 |
| 3 | 6.88 | 30.73 | 3.51 | 0.97 |
| 4 | 10.05 | 32.15 | 3.67 | 1.02 |
| 4.5 | 11.65 | 31.92 | 3.64 | 1.01 |
| 5 | 13.23 | 31.06 | 3.55 | 0.98 |
| 6 | 16.17 | 27.44 | 3.13 | 0.87 |
| 7 | 18.63 | 21.32 | 2.43 | 0.68 |
| 8 | 20.35 | 12.68 | 1.45 | 0.40 |
| 9 | 21.08 | 1.52 | 0.17 | 0.05 |

In order to provide more elements of the lagged response of climate processes, I emphasized the relative lag between South Pole (SP) and North Pole (NP) provided by the lags of jriT (2300yr) and GrT (3850yr), resulting in a 1550 yr lag of the NP respect the SP. This relative lag can be compared with previous works. For instance, a qualitative verification of these interhemispheric lagged responses is obtained from the S12 results, that, based on MonteCarlo simulations for the SH and NH temperatures (SHT and NHT), provide a set of leads and lags, with a mean lag (of NHT respect SHT) of around 1340 +/- 990 yrs. This relative lag is graphically shown in a comparison of the S12 evaluated SHT and NHT records with a maximum match obtained when the SHT is lagged of around 1400 yrs. It is shown in Figure 7b.

Motivated for the tidal recurrent influences, displayed in Figure 8a, the residual GT after to be rested the lagged TSI linear contribution (GT-m1GT[TSI]), is self-similar modelled. Figure 8b represents the non linear self-similar modelling (Eq. 3) that imply a forward temporal scale reduction a=0.96, and suggested a precession modulation, Mss (based on Eq. 4, and also displayed). This non-linear model, m2a, explains the last 18 millennia and forecasts the next four millennia. This non-radiative and non-linear residual, could be also explained with a gravitational model (Eq. 5) based on recently updated ephemris (NASA 2021b) for the Sun, Earth and Moon. The corresponding model m2b, based on lagged (by around 30 years) and linearly detrended and adjusted values of gravitational influences, is displayed in Figure 8c. The integration of results of all these GT modelling are displayed in Figure 9. It shows the GT record and the sum of their partial models. The sum of the trend and models m1 and m2a, constitute the deglacial process and the recurrent and lagged contibutions of solar and the tidal processes, that explains more than 95% of the global temperature variance.

Details of the forecasting results of these GT modelling are displayed in Figure 10. It integrates the trend, solar and the two tidal models (t, m1 and m2 and m2a) developed for the GT record. However, in Figure 10a, volcanic activity with increasing scalated values going downward from value 1.0°C, are depicted, indicating periods of volcanic activity with differences



beween tidal non-linear extrapolation and a tidal model based on ephemeris. For instance the period after AD 600 yr, where several volcanoes were strongly active.

In Figure 10b, a zoom of Fig. 10a, over the period 1500 2500 yr AD, is shown but with the increased contributions of solar and the tidal influences by 40% to consider both, the Maunder minimum of solar activity and the volcanic activity for the last
centuries, that increases, first a cooling and later, as a recovering processes, the warming of the 20[th] century.

Finally, in Figure 11, details of the multi-scale forecasting results of a TG modelling are displayed. Firstly an analog modelling of the residual GT values (GT-[m1GT(TSI)+m2bGT(TidalNAS)) are applied with a lag of 80 years (a recurrence presented by solar, typhoons and ENSO records [Ishizaki, 1971, Sánchez-Sesma, 2010]), an amplification of 1.4, and a bias of 0.25°C, that allows to extent its values along the 21[st] century. An explanation of variance of more than 59% (0.75 of
correlation coefficient) was obtained with this simple model. This amplified modelling, that could be considered conservative for its constant amplification, is depicted in Figure 11a. In the same Figure, a lagged (of around 66 yr) and linearly increased (0.45 ssn/yr) sunspot number record is also graphed (with dotted red line) in order to show the solar contribution at high frequencies and to demostrate the decaying lagged solar influences at the end of the 21[st] century. The sum of the multimillennia and self-similar multi-decadal scales modellings, constitute then a conservative GT scenario for
the 21[st]  century, considering the contributions of multi-scale climate approach. An important comparison for the 21[st] century of the GT instrumental record, the multi-scale modelling results, and the two lower mean scenarios of more than 45 models (IPCC, 2013), is depicted in Figure 11b.

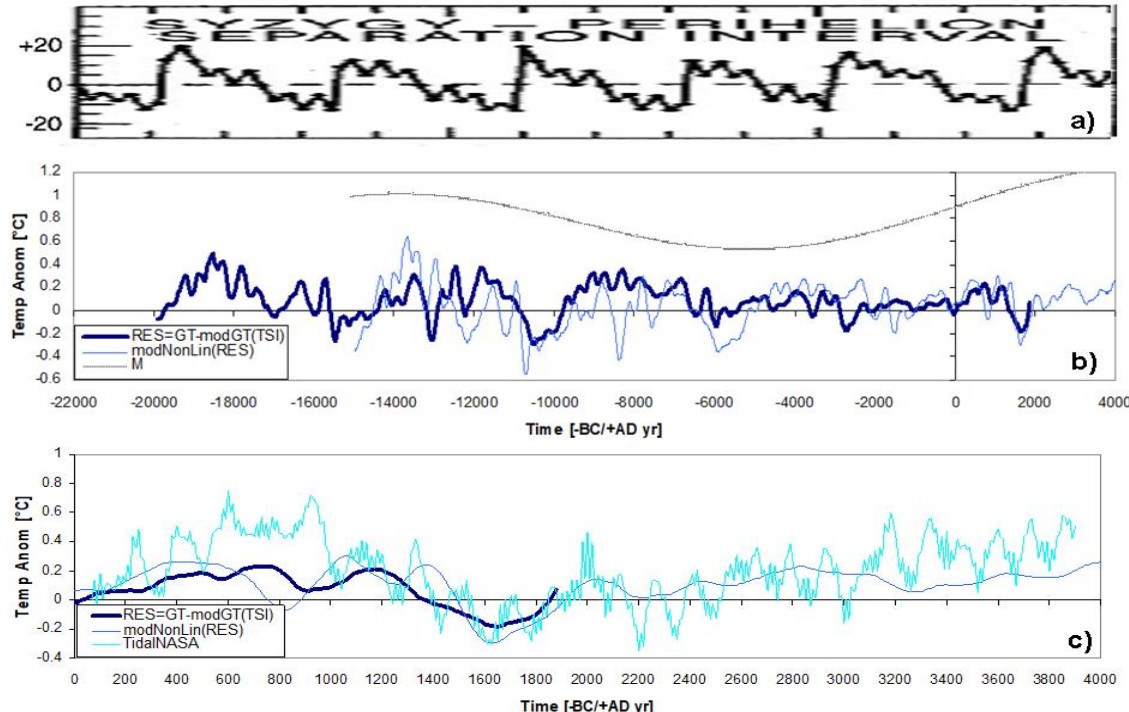

**Figure 8. Tidal influences and detrended global temperatures. a) Detrended non-solar global temperature (GT) component with its non-linear tidal model and forecasts; b) Non-linear tidal modelling of the self-similar non-solar GT component; c) a zoom of b) for the last and the next 2000 years, including the tidal results suggested by KW00 but based on NASA planetary information.**



**5 Discussion**

In this section we will discuss the obtained climatic results about recurrences, modulations, lagged responses, oceanic mechanisms and abrupt changes as a part of the proposed: Lagged & Recurrent Climate Hypothesis (LRCH).

The modelling of the climate records, justify two climatic characteristics: a) the solar recurrences (SS16) and b) the variable climate modulation (through the modulation factor $M_{IS}$). The modulation, of crbT and GT, depends on the ocean-atmosphere circulation, where glacial low (Holocene high) circulation imply high (low) thermal amplification. Then, the modelling of the polar and global temperature records, in terms of TSI exemplify the delayed and modulated actions of the OCB in climate processes.

The lagged responses to SF detected of around 80, 2300, 3850, 4070, 4450 yrs for the crbT, jriT, GrT, GT and samI, respectively, have shown, in a simple model, increasing lags for increasing accumulated OCB distance from the NE Pacific. In order to confirm the last four modelling results presented, they all together are integrated in a lag-accumulated_distance model. Using the Eq. 3, the input pairs of data, that are displayed in Table 1, are modelled (see Fig. 8) with a polynomial and the consequent very slow velocities, are shown in Table 2.

The proposed and applied lag-distance model, that provides a first validation of the LRCH, implies a well known oceanic mechanism, the Ocean Conveyor Belt (OCB), that is provided by the ocean near-surface streams, westward and poleward transports, of mass & heat, in a long-term journey, from the equatorial tropical Pacific to the Poles.

Other climate proxy reconstructions for hemispheric temperatures over the last 25 Kyr, provide elements for a second validation of the LRCH. [SH12 hemispheric temperature relative lags]

The results presented in this work, explained, as LR to SF for the crbT, jriT, GrT, GT and samI records, the following events and processes: a) the most of the Holocene climate variability, b) the major positive & negative changes happened during last millennia; and c) a consequent multi-millennial experimental forecast.

The existence of multiple contributions to GT variations is demonstrated by the modelled solar component, m1(TSI), and the tidal component, m2a(SS-NonLin) or m2b (TidalAcc). The value of these contributions is demonstrated, with variance

explanation (coefficient of correlation) between the global temperature and its complemented model [TG.vs.trend+m1(TSI)+m2(N-rad&N-Linear)] over the last 15 Kyr is 0.95 (0.975) (See Figure 9).

However, the most important result obtained here is the GT forecast for the next centuries and millennia. In particular, the GT cooling for the next decades is followed by a gentle warming displayed in Figure 10b. It is verified with independent information coming from the solar system NASA simulations that also indicated a cooling for the near future. It should be

also noted that, the multi-scale approach has allowed to decompose the GT variability into low and high frequency climatic processes. The high frequency component is empirically modelled as a self-similar and linear lagged model with 80 yr lag, based on Eq. 3. The 80 yr lag is detected both in solar and terrestrial records (Ishizaki, 1971; Sánchez-Sesma, 2010). The sum of these contributions shows a GT cooling for the next decades, in contrast with the lowest two IPCC GT warming climate scenarios (SPR-2.6 and SPR-4.5).

Although historical evidence reveals that natural occurrences of large, abrupt climatic changes are not uncommon (Alley 2000), they have occurred without any known causal ties to large radiative forcing change. Here we associate the abrupt complex responses of global climate (YD) event mainly to the lagged responses to solar forcing more than four thousand years later, and complementarily to the lagged responses to tidal forcing some decades later. This work, considers that all the most of the proposed possible mechanisms (amplifications, quasi-periodicity, an oscillation in deep ocean currents, "binge-

purge" cycle of ice sheets) could be associated with lagged solar and tidal influences. The LDEO researcher, Gerard Bond analyzed some abrupt events (Dansgaard–Oeschger events and Ice Raft Debris) and detected solar connections in the time and frequency domains, however, he did not consider the detected lags in this work. It should be emphasized that, during the YD event (included in the period from 11500 to 9000 yr BC) variance explanation (coefficient of correlation) between the global temperature and its models, proposed here, increase from 0.218 (0.467) to 0.76 (0.872) when the models consider

only TSI contributions [TG.vs.m1(TSI)] and the models consider both, TSI plus NonRad&NonLinear contributions [TG.vs.m1(TSI)+m2(N-rad&N-Linear)], respectively.

The conundrum of climate modelling associated with the middle holocene climate variations is explained here with the lagged solar and lunar influences on the GT records.





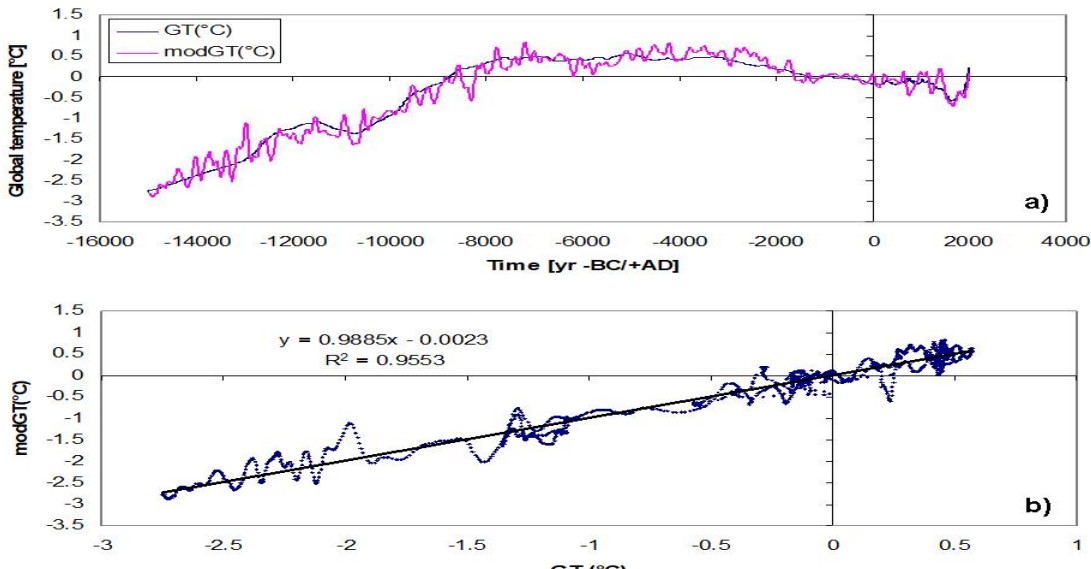

**Figure 9: Global temperature (GT) record and its modeling. a) The model sums the trend model, the TSI contribution and the non-linear tidal millennia-scale processes (see Figures 4d, 5d and 8b); b) the linear correlation and explained variance of the model.**

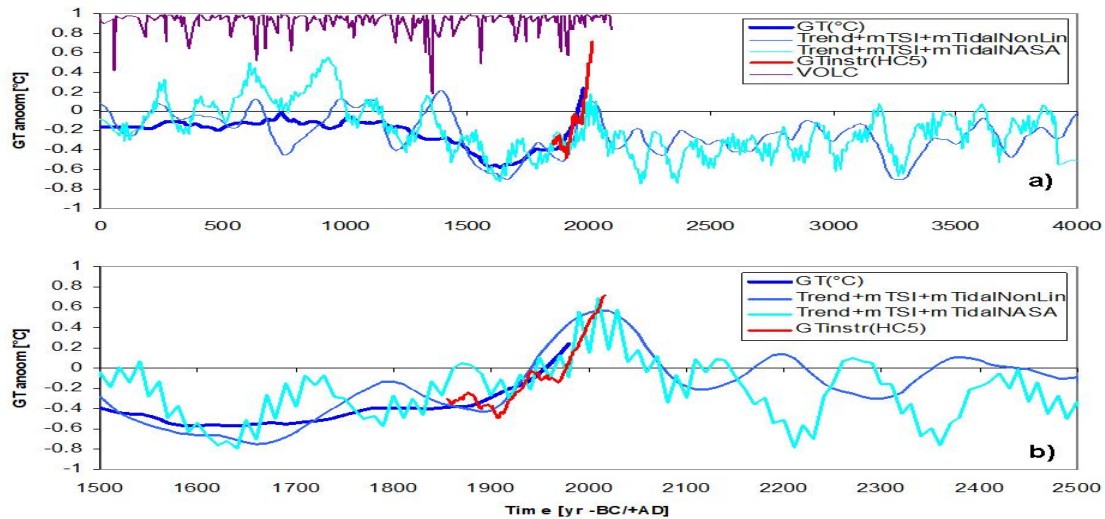

**Figure 10: Global temperature (GT) modeling and forecasts. The model based on the TSI records (SS16) and lunar tidal millennia-scale processes (see Figure 8). The HadCRUT5 instrumental record is displayed in red color.**



**6 Conclusions**

In this work, a Lagged & Recurrent Climate Hypothesis (LRCH) is proposed and discussed. This proposed LRCH, has presented, evaluated and discussed elements to develop a third step of contributions in the "long-road", initiated almost 100 years ago, required to develop, a complete Multi-scale climate modelling (MSCM). The first contribution, associated with orbital influences, has provided elements to consider the relatively low-frequency oscillations (with periods going from $10^4$ to $10^6$ years) climate recurrent and lagged responses to planetary forcing to the Earth´s dynamics. A second contribution, has been provided by the "traditional" climate approach developed since 1950s that includes annual, multi-annual and decadal climate variabilities, with its relatively high-frequency oscillations (with periods going from 10 to $10^2$ years).

In this work, the LRCH, can be considered a third contribution, associated with millenia-scale oscillations fueled mainly by solar and lunar influences. The multi-millennial oscillations cover the great "hole", left between the "orbital" and "traditional" climate approach (with periods going from $10^2$ to $10^4$ years).

This proposed LRCH is based on the recently reconstructed SF and EC records over the last deglaciation and the Holocene. As climate research must be based on both climate records and known physical processes, our research work proposed an hypothesis that relates solar activity and delayed climate responses to oceanic mass and heat transport. As our hypothesis contributes to explain the climatic variations of several of the most important terrestrial processes, we consider this hypothesis as a scientific one.

Similarly, to all geophysical and thermo-mechanical systems, the climate system processes have presented lagged responses (LR). These EC-LRs have been documented thanks to the glacial-interglacial reconstructions of climate processes developed during the last decades. For instance, S12, have analyzed the 23, 41 and 100 Kyr with climate LR of around 3, 5 and 20 Kyr, respectively. Additionally, a shorter forcing oscillation, a multi-millennia solar recurrence, has been recently detected [SS16].

In this work, these kind of lagged responses are considered an important part of a climate response process to SF. These LRs can be assessed taking into account recent reconstructions of SF and EC events over the last millennia. Moreover, the main mechanism for the linear LR of the EC is suggested, it is the well oceanic processes: "ocean conveyor belt" (OCB) [Broecker (1991)]. In this work, it was shown how the OCB influences generates delayed climate responses, however, as NOAA (2021b) has pointed out, there are other important characteristics of the OCB: "It is also a vital component of the global ocean nutrient and carbon dioxide cycles. Warm surface waters are depleted of nutrients and carbon dioxide, but they are enriched again as they travel through the conveyor belt as deep or bottom layers. The base of the world's food chain depends on the cool, nutrient-rich waters that support the growth of algae and seaweed."

In contrast with the conventional thoughts about OCB that supposedly began on the surface of the North Atlantic ocean near the pole, in this work it is considered that the OCB begins on the surface of the Eastern tropical Pacific waters near the Equator where the solar radiation (and its influences) is at its maximum value. Here, the water is vertically mixed by trade currents through the Ekman pumping. This middle-deep water moves west, at the Equator, passing all the Equatorial Pacific, toward the Indian, and going south to the ends of Africa and turning East around the edge of Antarctica, where the winds are stronger and sustained, causing the conveyor belt to increase velocities, or be "recharged." After it moves around Antarctica, the conveyor turns northward going into South Atlantic East coast and after passing the Equator, going into North Atlantic West coast. This surface, or first stage, of the OCB is fundamental to explain the climate lagged links between different zonal processes.

In addition, in this work, solar-lunar influences, were evaluated twice, through: a) a self-similar non-linear recurrent processes, and b) a numerical gravitational calculation based on NASA´s planetary positions over the last and future millennia. This second evaluation is very important, because it provides elements not only to verifiy the non-linear evaluated tidal influences, but also to verify the correctnesses of the hypothesized solar lagged influences. The NASA horizon tool (NASA 2021b), provides solar, lunar and Earth, positions and velocities over the last and future millennia that support the most objective analysis of a tidal component, resulting in an important linear contribution to GT variability.

The outcome of this work explains as EC-LR to SF and LF, the bulk of the Holocene EC variability and a consequent experimental multi-millennial forecast of the EC for the next centuries. In particular the GT forcast shows a cooling trend for the next decades, centuries and millennia.

This work modelling has shown two main multi-millennial scale capacities. One is associated with the past climate variability, because this work has presented and discussed a possibly solar lagged forcing of the YD event, considering





**Figure 11: Global temperature (GT) multi-scale modeling and forecast. a) The residual of the GT instrumental record (HadCRUT5) and its model based on the TSI records (SS16) and lunar tidal millennia-scale processes (see Figure 10) is self-similar modelled wih a linear transformation and a lag of 80 years. The lagged (of around 66 yr) and linearly increased (0.45 ssn/yr) sunspot number record is also graphed (with dotted red line) in order to show the multi-scale solar lagged influences with SSN monthly record (SILSO 2021); b) The GT instrumental record (HadCRUT5) is compared with the sum of two models Multi-millenial + Multi-Decadal (shown in Fig 11a) and the two lower scenarios RCP2.6 and RCP4.5 of IPCC 2013 (See comments in the text).**





the lag time of around 4 Kyr that requires the movement of the tropical waters to reach the Arctic. The other is associated. with consequent millennia-scale forecasts that, after being verified with the Lag-AccDist model, indicates gentle oscillations for the next millennia.

5 However, an experimental application of the multi-scale approach has permitted to decompose the 21$^{st}$ century GT variability in two processes, one associated with millennia scale solar and lunar influences, and other associated with multidecadal recurrent processes, that suggest a cooling for the next decades

Final remarks:

- During the last decade, long-term climate responses, mainly associated with sea-level, have been indicated by the most recent models; however, these long-term (or lagged) responses are "focused" to magnify the GW impacts over the next millennia. In contrast, in this work these lagged responses (due to ocean mass and heat movements) have been analyzed, synthesized and employed, firstly, to explain the past millennia variabilities, and secondly, to suggest their future trends and variations.

- Although S12 has pointed out that: "global warming preceded by increasing carbon dioxide concentrations during the last deglaciation", and the results of this work also indicate that $CO_2$ preceded GT variations, however both records could be considered as lagged responses, several hundreths of years later, of solar activity. As S12 has demonstrated the Antarctic climate variability, changes almost in synchrony with atmospheric carbon dioxide variability, implying that $CO_2$ lags solar activity by more than 2000 years. Then the carbon dioxide, appears to follow, many centuries later, solar activity as a part of oceanic biochemical complex dynamics in the Equatorial

Pacific. As Clement et al. (2001) have pointed out: "changes in the tropical climate are known to have global impacts on interannual timescales and are therefore a likely candidate for generating global climate change on longer timescales."

- Only in the cases in which a proposed model explains the processes more fully, can it be considered a scientific hypothesis. Therefore, the hypothesis proposed in this work, although empirical, can be considered to constitute a

scientific climate hypothesis. However, only after being reaffirmed by subsequent independent verifications and theoretical investigations would the proposed hypothesis would become a scientific theory.

**Acknowledgements**

This study is dedicated to the following climate researchers: Rodhes Fairbridge, John Sanders, Wallace Broecker, Gerard Bond, George Kukla, Walter Munk, Charles Keeling and Thomas Whorf, who with their works and words motivated the

author to detect, model and better understand the solar and lunar, lagged and recurent, influences on climate. Many thanks to the teams of researchers devoted to reconstructing solar climate records such as those led by Solanki and Steinhilber, and to exploring tidal influences on climate leaded by Keeling. At last, but not least, many thanks to the teams of researchers focused on reconstructing proxy and instrumental climate records led by important researchers exemplified by: Wiejer, Mulvaney, Alley, Marcott, Shakun, Wang, Moberg, Neukom, Hansen and Jones. The author also wishes to thank the kind

comments and suggestion made by Susan Aliphat, Javier Zavala-Garay, Jaime Whaley and Francisco José Sánchez-Sesma to the first versions of this work.

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



## Appendix A. Calibration of the Global Temperature reconstructed record (Marcott et al 2013).

A simple model to calibrate climate records is proposed. A linear and lagged model of $C_I(t)$ in terms of $C_J(t)$, is defined as:

$$C_I(t) = \alpha_{IJ} C_J(t - \delta_{IJ}) + \beta_{IJ}(t - t_1) + \gamma_{IJ} + e_{IJ}(t), \tag{A1}$$

Here, α are the amplification factors, β are the slopes, δ are the lags, $\gamma$ are the additive constants, $t_1$ are the initial times for the modeled period, and $e(t)$ are the analogue error of these models. The first and second subindexes, I and J, indicate the reconstructed record numbers, to be modelled and used as a base to, respectively.

The adjustment of the global temperature records is carried out in two stages. In the first stage, at the hemispheric level, the NHT and SHT values are adjusted or calibrated (in time and values) with the respective instrumental records HadCRUT5 (HC5); These processes are shown in Figures A1a and A1b. In the second stage, with the ajusted hemispheric records, NHTa and SHTa, they are averaged and the global record is calculated; it is shown in Figure A2. The final stage of the calibration processes is also depicted in Figure A3.

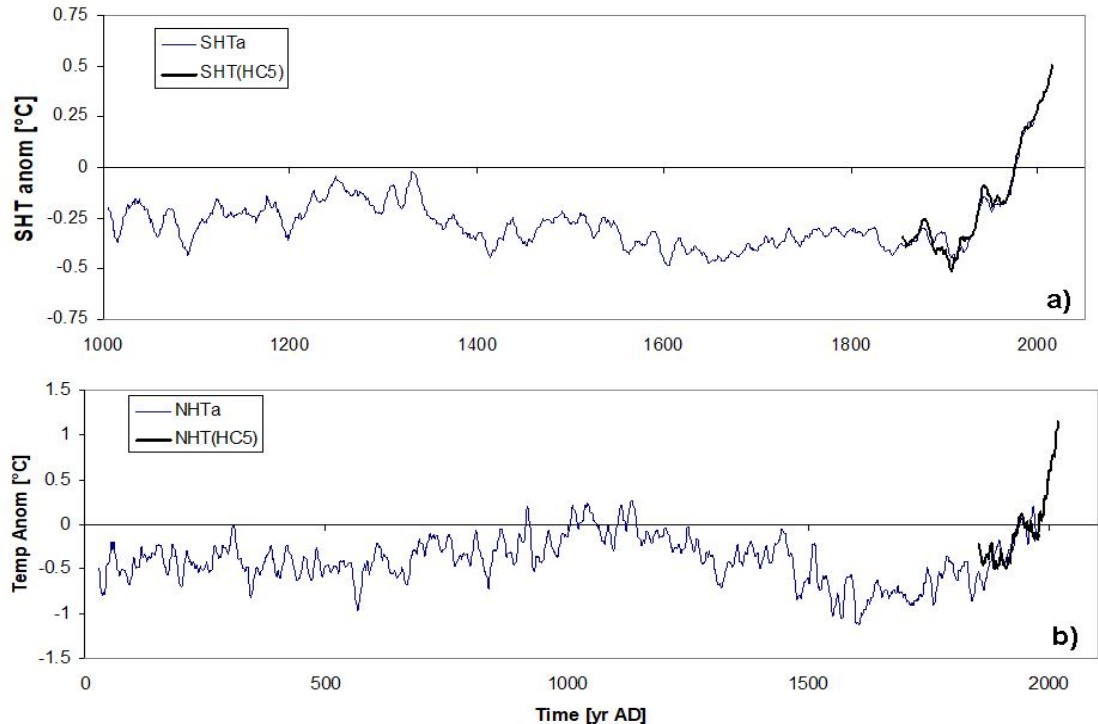

**Figure A1: Adjustment of Hemispheric temperature reconstructed records to instrumental records. Adjustment of a) Southern hemisphere records (N13) and b) Northern hemisphere record (M05) with the instrumental records (HC5). The parameters are: 0 yr lag, 1.35 amplif, 0.0 bias, and NHT 24 yr lag, 1.4 amplif, 0.1 bias for the SHT and NHT, respectively.**



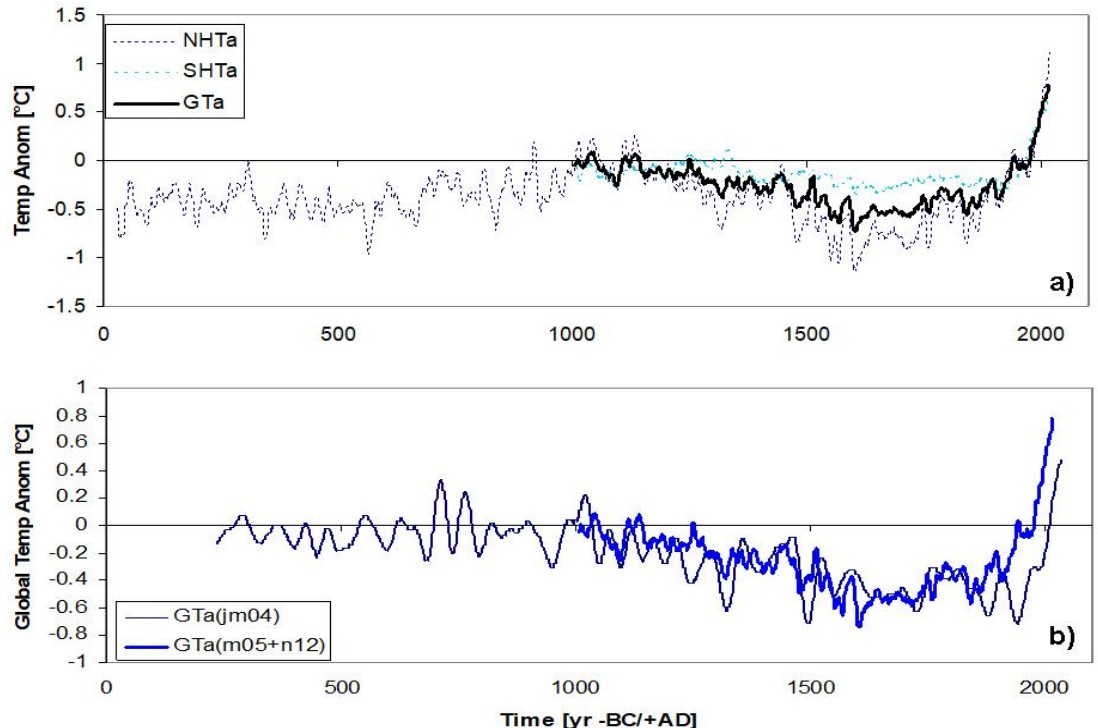

**Figure A2: Global temperature adjustment. a) Integration of a global record for the hemispheric adjusted records (see Fig. A1); b) the global temparture record from S12 and M13 is calibrated with the GT record depicted in a).**





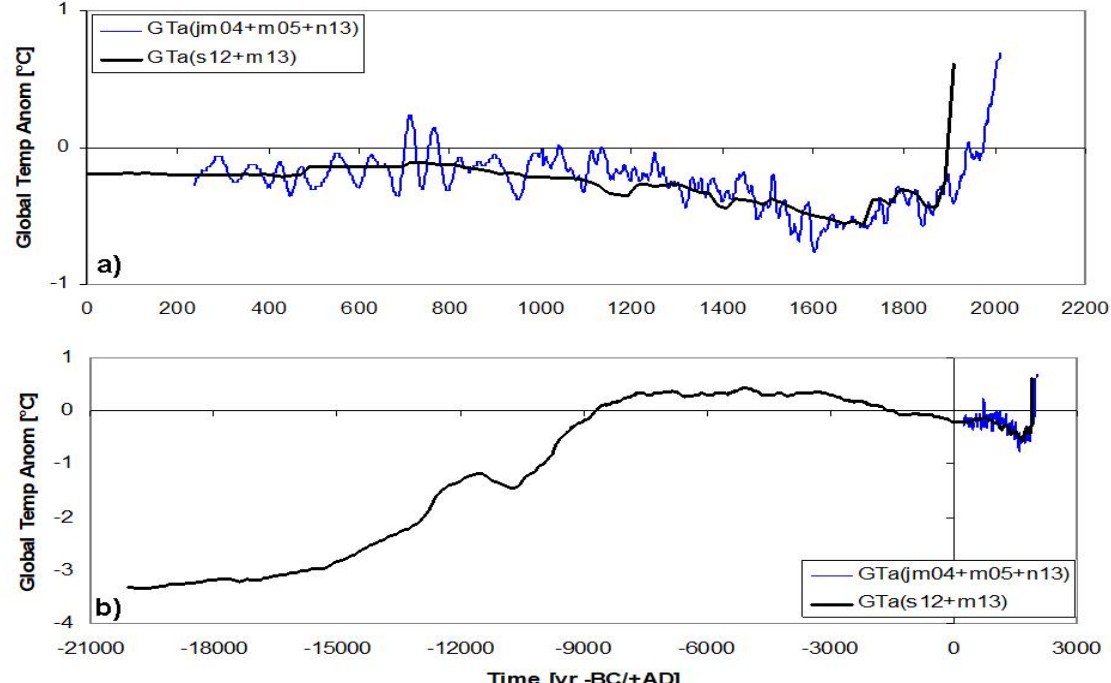

**Figure A3: Global temperature adjustment. a) Integration of a global record for the hemispheric adjusted records (see Fig. A1); b) the global temparture record from S12 and M13 is calibrated with the GT record depicted in a).**





## Appendix B. Climate and Solar lagged correlations and common recurrences

The detection of the optimum lag of solar influences to explain the five main climate records are shown in the following five Figures. Also a Fourier analysis show the 9500 yr recurrences of both TSI and the five analyzed records.

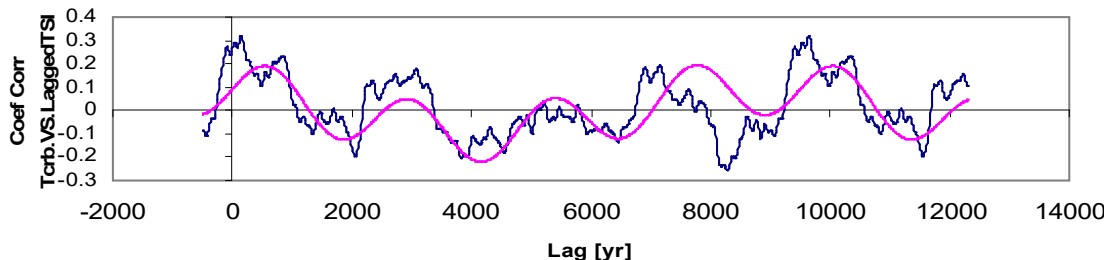

**Figure B1: Correlation analysis for the Congo River Basin temperature (crbT) (Weijer et al., 2007) lagged responses to TSI (SS16). A continuos line (cyan color) indicates a recurrent Fourier Series (FS) function (and its second harmonic) with a period of 9500 yrs. A correlation coefficient of 0.603 between the lagged correlation and the adjusted FS, indicate the lagged and recurrent solar influences on the crbT record.**

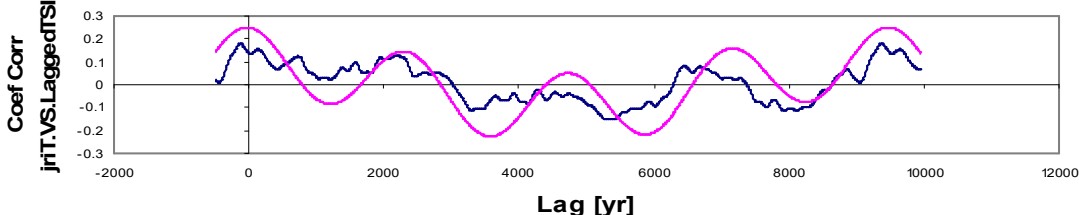

**Figure B2: Correlation analysis for the antarctic James Ross Island temperarture record (jriT) (Mulvaney et al., 2012) lagged responses to TSI (SS16). A continuos line (cyan color) indicates a recurrent Fourier function (and its second harmonic) with a period of 9500 yrs. A correlation coefficient of 0.745 between the lagged correlation and the adjusted FS, indicate the lagged and recurrent solar influences on the jriT record.**

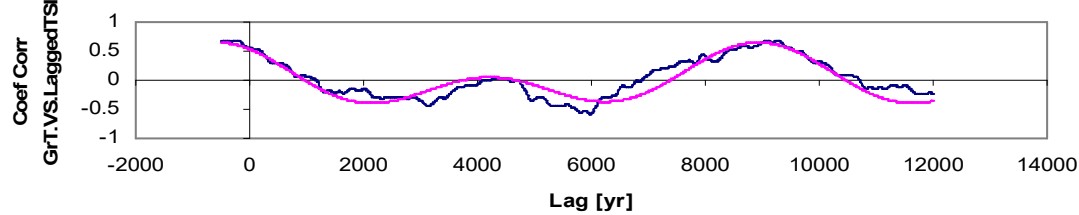

**Figure B3: Correlation analysis for the Greenland temperarture record (GrT) (Alley et al., 2000) lagged responses to TSI (SS16). A continuos line (cyan color) indicates a recurrent Fourier function (and its first harmonic) with a period of 9500 yrs. A**




correlation coefficient of 0.916 between the lagged correlation and the adjusted FS, indicate the lagged and recurrent solar influences on the GrT record.

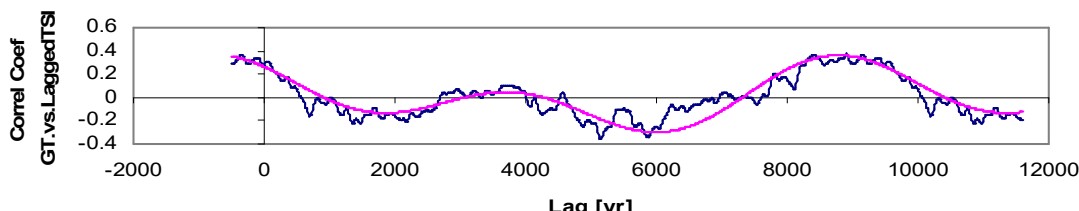

Figure B4: Correlation analysis for the Global temperature (Shakun et al., 2012; Marcott et al., 2013) lagged responses to TSI
(SS16). A continuos line (cyan color) indicates a recurrent Fourier function (and its first harmonic) with a period of 9500 yrs. A correlation coefficient of 0.907 between the lagged correlation and the adjusted FS, indicate the lagged and recurrent solar influences on the GT record.

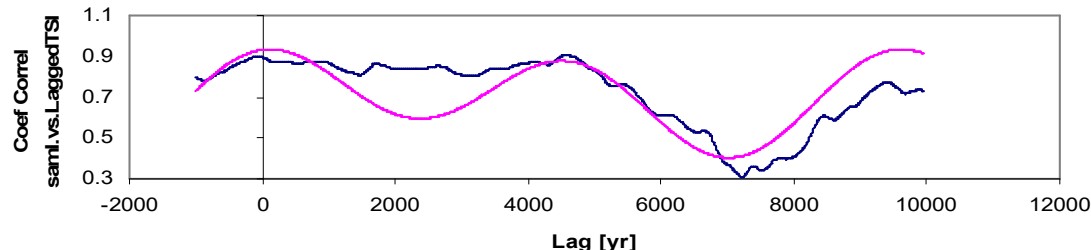

Figure B5: Correlation analysis for the Dongee cave Isotopic Index (Wang et al., 2000) lagged responses to TSI (SS16). A continuos
line (cyan color) indicates a recurrent Fourier function (and its first harmonic) with a period of 9500 yrs. A correlation coefficient of 0.732 between the lagged correlation and the adjusted FS, indicate the lagged and recurrent solar influences on the samI record

For the five models [climate(laggedTSI)] presented, the selected lags correspond [applying Pearson algorithm (Numerical recipes, 1997)] to the following: a) Corr. Coef.: 0.459625E+00, 0.510491E+00, 0.232874E+00, 0.474634E+00, and 0.928824E+00, respectively; b) Probability:  0.000000E+00, 0.000000E+00, 0.151949E-37, 0.000000E+00 and
15 0.000000E+00, respectively.

