# Peer review of "The Earth's climate lagged, recurrent and non-linear solar and lunar multi-millennial scale responses: An oceanic hypothesis, evidence, verifications and forecasts"

_Earth System Dynamics, 2021_

## Referee Comment (RC1)

Review of the manuscript 'The Earth's climate lagged, recurrent and non-linear solar and lunar multi-millennial scale responses: An oceanic hypothesis, evidence, verifications and forecasts' by Jorge Sánchez-Sesma (Independent researcher).

The author presents the 'Lagged & Recurrent Climate Hypothesis' (LRCH), claiming that much of the climate millennial scale variability (with periods between some hundreds of years and a few millennia) is driven by intrinsic solar variability and extreme tides associated with Sun-Earth-Moon orbital coincidences, reoccurring at certain repeated periods. Those extreme tides generate cooling of the sea surface by increased vertical mixing. The lagged response of that forcing in the surface temperature is driven by the Ocean conveyor belt (OCB) with the lag being the 'age of the water' roughly measured by the 'oceanic accumulated distance from the tropical East Pacific'.

The above mechanistic theory is quite simplistic and even not falsifiable according to a fundamental Popper's criterium (https://en.wikipedia.org/wiki/Falsifiability) for the validity of scientific theories, The justification of the theory is based on: 1) ad-hoc, physically not well-grounded arguments, generally taken from ancient literature and 2) a set of (non-statistically validated) fitting regressions using over-simplistic linear modulated/lagged models using as predictors extrapolated data by Fourier analysis of the total solar irradiance (TSI) at every 9.5kyr and of the tidal forcing at every 5kyr. The 9.5kyr solar recurrence is misleadingly supported by the same author in Sesma 2016 (SS16 in the manuscript) by a spectral analysis without any statistical significance study analysis of the spectral peaks. That replication (both in the past and future) for a multiple number of Fourier analysis time-series length (9.5 Kyr) is extremely dangerous due to the high dependence of phases of Fourier components from the analysis period. Moreover, the concept of 'age of water' is quite ambiguous because the OCB is a graph with multiple bifurcations, dispersion, stochasticity etc., hence not allowing to estimate the age of water by simple backward cinematics.

The author commits too many methodological and technical flaws in his presentation and discussion of results.

Moreover, the author does not cite or contextualize in his research, related (well supported and methodologically much more advanced) works studying the possible origins of the millennial variability at scales slower than that of Milankovitch forcings. In fact, it is an open discussion, far from being at the end, if the above referred variability comes mostly from the solar/tidal forcing or from the internal slow (not externally forced) variability of the climate system (Soon et al. 2016) or even from magnetic forcing.

The author's belief both in the LRCH and his simplistic model, led him to produce forecasts of the global surface temperature for the next decades, in a clear confrontation and contradiction with the decadal forecasts of the much more sophisticated and physically well-grounded models presented worldwide in the IPCC reports.

Despite the author's recognition in the conclusion of the paper, that his model deserves much more work and physical evaluation, the present work cannot be considered appropriate to be published in the scientific journal ESD.

Next, there are presented some of the major technical and methodological pitfalls of the manuscript:

1) Figure 1a is totally inappropriate
2) Figs 3a and 3b are not legible. The adapted Fig. 3c is too much stretched and without legibility.
3) The model in Eq. (1) of the climate proxy variable, uses the solar proxy variable S(t) which is supposed itself to be self-similar (with a 9.5kyr period). Therefore, any slower variability with periods larger than 9.5kyr is ignored. The regression score (e.g. MSE) of the fitting is never shown systematically for the tested climatic proxies neither compared against any 95%-expected score of a null-hypothesis model (e.g. an red noise AR1 model). Therefore, the absence of statistical validation is not allowing any attribution arguments of the variability to the tested external forcing.
4) Eq. (5) is completely out of the context and add nothing about the tidal forcing particularities.
5) The details of models (Section 4) should preferentially be presented in a Table.
6) Fig.4 is not clear at all. All panels (except panel b) present two curves which are not explained anywhere, even in the caption. Observed proxy data of some of the time-series does not match perfectly with graphs of the papers from which they where obtained. For instance: a) the Record crbT (green curve of fig 4a) seems to be a degraded representation of data presented in Fig. 2c of Weijers, *et al.(2007). b) Record jriT (fig. 4b)* representing the Northeast Antarctic Peninsula temperature, jriT (Mulvaney et al., 2012) should represent in effect an anomaly (with respect to the period 1961-1990) in the Antarctic James Ross Island, presented in Fig. 3 (black curve) of Mulvaney et al., 2012. The curves don't match.
7) The Figs 4d and 5d, supposedly to present the global temperature anomaly do not match. Difference between curves are not explained anywhere.
8) Fig. 8a is an appropriate stretched copy-paste figure without any information of the time scale.

9) Figs. 8b and 8c presumably showing the contributions of the solar and tidal influences to the global temperature, are not all well explained. The explained variances are not computed and tested against null hypothesis.

10) The manuscript is full of confusing acronyms, which are not well defined in the text, like NonRad&NonLinear , TidalNonLin , climate(laggedTSI).

11) The description of the tidal contributions in the Discussion section 5, pg. 13 lines 23-34 is quite confusing.

12) Figs 10a,b present forecasts of the global temperature up to 4000 cal A.D. The author claims that the use of the NASA's solar system astronomical model represents an independent verification of the model. For instance, the time-series issued from models Trend + mTSI + TidalNonLin (blue line) and Trend + mTSI + TidalNASA (cyan line) appear to be quite different and hence not corroborating the empirical proposed model Trend + mTSI + TidalNonLin.

References

Soon, Willie; Velasco Herrera, Victor M.; Selvaraj, Kandasamy; Traversi, Rita; Usoskin, Ilya; Chen, Chen-Tung Arthur; Lou, Jiann-Yuh; Kao, Shuh-Ji; Carter, Robert M.; Pipin, Valery; Severi, Mirko; Becagli, Silvia (2014). *A review of Holocene solar-linked climatic variation on centennial to millennial timescales: Physical processes, interpretative frameworks and a new multiple cross-wavelet transform algorithm. Earth-Science Reviews, 134(), 1–15.* doi:10.1016/j.earscirev.2014.03.003

---

## Author Comment (AC1)

**AUTHOR´S COMMENT IN RESPONSE TO ESD-2021-84-RC1.**

Cuernavaca, Morelos, Mexico January 14[th], 2022.

Dear paleoclimate colleagues:

It is for me a great opportunity to comment (with discussions and additional verifications) my work (esd-2021-84) that presents several important and objective elements of the lagged and recurrent climate hypothesis (LRCH) that could support the development of a consequent (novel & needed) scientific theory of climate. Although the proposed LRCH could not be considered a scientific hypothesis, it provides several important elements to develop, first a scientific hypothesis, and later, a scientific theory.

Taking into account the anonymous reviewer comment (RC1), it is possible to extend and better define a supporting argumentation (written and graphically) of my work (esd-2021-84). Although, the RC1, after to argue about a supposed theory proposed, pointed out: "the present work cannot be considered appropriate to be published in the scientific journal ESD", let me say that: *I do not agree with this veredict*.

The written argumentation, to support my work (esd-2021-84), is the following:

1. Earth´s climate (EC) processes are complex. The EC is linked not only to terrestrial processes, but also to cosmic processes. Then, EC should be considered an open process, that, at least, requires the amplest spatial and temporal frames of references for solar and lunar millenial scale influences. [Note: The orbital (or planetary) forcing of climate, devoted to oscillations greater than 20 Kyr, that were proposed by Milankovitch (1931) required more than three decades to be accepted].

2. In order to disentangle the climate complexity, firstly, I have developed a work about a multi-millenia scale solar recurrent pattern (Sánchez-Sesma, 2016; hereafter SS16). This paper published in 2016 in ESD, detects, through Fourier Analysis, a recurrent pattern. This recurrence was empirically validated both, with independent isotopic and climate records ( in quantity and quality), and with a gravitational model based on objective solar system dynamics (Horizon system/JPL/NASA). Although this published work has been referred by several works (vgr;Viaggi et al., 2021), however it is possible to present additional verifications (see them below).

3. Secondly, in my work ESD-2021-84, I have developed an integrated empirical explanations of some climate processes, over the last millennia. These explanations are based both on the solar recurrences detected and also on their lagged influences associated with ocean circulation. The well-known (but not fully applied in climate) "ocean conveyor belt" (OCB) concept was considered as the main part of my work. In this comment I have presented additional evidences (see them below for different scales and different oceans).

Previously to present my detailed argumentation, and to mention that all the corrections related to the RC1 document, will be considered in the final version of this work, I would like to make a final comment: "*This work is about three geophysical science processes. One is the multi-millennial potential planetary forcing of the Sun, which generates a quasi-deterministic recurrences that, as the orbital forcing which generates recurrent changes in the Earth's orbit, also generates (by an unknown mechanism) recurrent solar forcing. Another is the lagged responses, characteristic not only in mechanical and thermal processes, but also in climate processes, where the orbital influences generate lagged responses. And another is about the development of geophysical sciences, that required long-term processes to update and change accepted models.*"

In order to support my written argumentation, additional evidence that support the LRCH is presented. In the following detailed information, analysis and/or explanations are provided:

a) The existence of recurrence in solar activity was analyzed in a specific paper at ESD. However there are several important aspects to be remarked. In SS16 there are: 1) a recurrent gravitational possible forcing that suggests the generation of solar activity long-term changes, 2) several verifications that the solar recurrent pattern persists since geological eras. Additionally, there are recent studies that provide more reconstructions (Vaggio, 2021).

b) A muti-scale (geological, millennial, decadal, and annual) analysis of the lagged climate processes in the Equatorial Pacific, and a detailed lagged processes between Equatorial sea surface temperatures between Pacific & Indian oceans over the last centuries.

c) Two additional verifications of the lagged solar influences, one for the Arctic and another for the tropical climate.

d) An additional GT scenario for the 21$^{st}$ century is presented. It is based on a Northern Sweden isotopic record that leads global temperature by more than 250 years.

Finally, as my work provide elements to support another climate explanation than that provided by the the current "anthropogenic global warming" (AGW), it is important to emphasize three simple facts, that question the AGW theory (see below). These facts are about the relative occurrence of $CO_2$: a) One covering the last 50 Kyr, shows the relative leads of $CO_2$ respect Antarctic climate, that changes to relative lags when volcanic activity increases since the last glacial maximum; b) Other covering the Holocene, shows the $CO_2$ variations explained as a lagged response to solar variations; and c) Other covering the last decades, shows the monthly occurrences of $CO_2$ changes after its solar forcing occurrences.

**A. Verification of solar recurrences (with a pattern of 9.5 Ky) and their influences on climate**

Several verifications of solar recurrences are presented here.

1) The last geomagnetic reversal (GMR) has been an interesting event. Ueno et al., 2019 have collected and integrated climate information in order to analyze the associated anomalies caused by this GMR event. Figures A1 and A2 show comparisons of extrapolated TSI (SS16) with two climate proxy records, along 19 and 18 Kyr, from NW Pacific and N Atlantic, respectively. These figures show excellent matches with the TSI extrapolated around 80 times 9.5 Kyr back in time.

[Figure]

**Figure A1. Comparisons of extrapolated TSI (SS16) with a climate proxy record of biogenic productivity from NW Pacific ocean [Ueno et al., 2019].**

[Figure]

**Figure A2. Comparisons of extrapolated TSI (SS16) with a climate proxy record of SST from N Atlantic ocean [Ueneo, et al., 2019].**

2) Another comparison is based on the reconstructed Greenland temperatures (Clark et al., 2020) for the NEEM (Greenland) ice-core site (dark blue line, gray shading is uncertainty). The comparison with the TSI (SS16) extrapolated back in time (13 times of 9.5 Kyr) let us estimate a lag of around 3900 years.

[Figure]

**Figure A3. Solar pattern explaining Greenland temperatures (GrT) over the last interglacial (Clark et al., 2022). Solar pattern of 9.5Kyr with a temporal time of 13400 yr BP, that matches GrT at 123.5 Kya (13 times 9.5 Kya) is implying a GrT lag, respect TSI, of around 3.9 Kyr.**

3) Another important set of independent evidences for the TSI recurrences of around 9.5 Kyr, is developed byViaggi et al. (2021). They have presented a comparison of sub-Milankovitch cycles (in kyr) identified in the EPICA records by the singular spectrum analysis (SSA) with other signals and their interpretation (forcing) from literature. In Table 1, the part of his table showing periods between 8.8 and 10 kyr (for 7 signals [EPICA δD, EPICA CO2, EPICA CH4, Indian Ocean δ18O, Rhythmites cyclostratigraphy, Rhythmites (coupled thickness and magnetostratigraphy), Rock magnetic cyclostratigraphy] and 3 forcings [Planetary beat on Sun, Nonlinear Milankovitch oscillator, Combination tones of Milankovitch cycles]), is the following:

**Table 1. Comparison of Sub-Milankovitch cycles detected in Antarctic ice-cores and other parts of the world, and their estimated forcing (recorted from Viaggi [2021])**

| SIGNAL | AGE | SUB-MILANKOVITCH CYCLE | | |
|---|---|---|---|---|
| EPICA δD m | Middle Pleistocene-Holocene | 13 | 9.2 | 5.7 |
| EPICA CO2 m | Middle Pleistocene-Holocene | 13 | 9.8 | 5.9 |
| EPICA CH4 m | Middle Pleistocene-Holocene | 13 | 9.0 | 5.4 |
| Indian Ocean δ18O f | Late Pleistocene | 12.8–13 | 9.1–9.3 | 5.6–5.9 |
| Rhythmites cyclostratigraphy k | Paleozoic | 11.0–12.7 | 8.9–9.2 | 4.7–6.0 |
| Rhythmites (coupled thickness and magnetostratigraphy) l | Late Carboniferous-Early Permian | 11.1–13.6 | 8.8–10 | 5.1–6.2 |
| Rock magnetic cyclostratigraphy g | Early Triassic | 12.3–12.6 | 8.9–10.0 | 4.7–5.6 |
| **FORCING** | **AGE** | **SUB-MILANKOVITCH CYCLE** | | |
| Planetary beat on Sun | – | – | 9.6 **a** | 4–5 **n** |
| Nonlinear Milankovitch oscillator f | – | 12.4–13 | 9.0–9.5 | 5.5–5.8 |
| Combination tones of Milankovitch cycles | – | 10–12 **h** | 9.2 **l** | 6–7 **h** |

NOTES
**a** Sánchez-Sesma (2015, 2016).
**b** Dergachev (2004); Xapsos and Burke (2009); present study from data of Vieira et al. (2011) and Steinhilber et al. (2012); Usoskin et al. (2016); Usoskin (2017).
**c** Creer and Tucholka (1983); Hagee and Olson (1989); Gogorza et al. (2000).
**d** Viaggi (2018).
**e** Rodrìguez-Tovar and Pardo-Igùzquiza (2003).
**f** Pestiaux et al. (1988).
**g** Wu et al. (2012).
**h** Da Silva et al. (2018).
**i** Mayewski et al. (1997).
**j** Olsen and Hammer (2005).
**k** Elrick and Hinnov (2007).
**l** Franco and Hinnov (2012); Franco et al. (2012).
**m** This work (Viaggi, 2021)

It is important to mention that the average value of the periods analyzed by Viaggi (2021), shown in Table 1, is 9.35 Kyr.

4) Another qualitative comparison is developed with 14C isotopic anomalies measured from Greenland ice cores by Lal et al. (2005). The authors of this work, have pointed out: "The observed variation in cosmic ray flux at the polar site is best attributed to changes in solar activity resulting in variable modulation of terrestrial cosmic ray flux." The comparison with the extrapolated TSI (SS16), and the 14C anomalies after to be adjusted both through a linear transformation and a bias of 1800 yrs, is shown in Figure A4 with a very good qualitative agreement.

[Figure]

**Figure A4. A comparison of 14C local production anomalies measured from Greenland (Lal et al., 2005), with the extrapolated TSI (SS16). The results of 14C after to be linearly transformed and a bias of 1800 yrs, shows a qualitative agreement between these records.**

**B. Comparison of phase lags between west and east Equatorial Pacific SST, in the geological, millennial, multidecadal, and annual scales; and a detailed analysis of lags between coral reconstructed records from east Equatorial Pacific, west Equatorial Pacific and west Indian, over the last four centuries.**

Four forcing periods were considered: 9 Myr, 9.5 Kyr, 80 and 1 yrs. The corresponding graphical analysis are displayed in Figures B1- B4.

For the geological scales, based on Fox et al. (2000), a 0.8 Myr lag is estimated between west Equatorial Pacific SST and east Equatorial Pacific SST. It is shown in a comparison of SST for east and west Eq. Pac. It is shown in Figure B1.

[Figure]

**Figure B1**

For the millennial scales, based on Lea et al., 2000 and Lea et al., 2006, a 0.9 Kyr lag is estimated between west Equatorial Pacific SST and east Equatorial Pacific SST. It is shown in Figure B2.

[Figure]

**Figure B2**

For the decadal scales, based on the NOAA´s data, a lag of 12 years is estimated between Niño 4 in the west Equatorial Pacific SST and Niño 12 in the east Equatorial Pacific SST. It is shown in Figure B3.

[Figure]

**Figure B3**

For the annual scales, based on the NOAA´s data, a lag of 3.5 months is estimated between Niño 4 in the west Equatorial Pacific SST and Niño 12 in the east Equatorial Pacific SST. It is shown in Figure B4.

[Figure]

**Figure B4**

[Figure]

**Figure B5: Ratio between West Pac SST lags respect East Pac SST and their forcing periods. Forcing periods were four 9 Myr, 9.5 Kyr, 80 and 1 yrs. See the four correspondent graphical analysis in Figures A1- A4.**

Tierney et al. (2015), integrated SST reconstructions based on coral information coming from East Pacific, West Pacific and Indian Oceans, over the last four centuries. These records and their relative lags are displayed in Figure B6. The detrended correlations show their maxima, of smoothed values, for Epac lags of around 7 and 13.5 yr, for best matches for Wpac and Indian records, respectively.

[Figure]

**a)**

[Figure]

**b)**

**Figure B6: Cross-correlation analysis of east and west Pacific, and west Indian SST records, with different lags over the last centuries. a) records from Tierney et al., 2015; b) detrended correlation for different lags of the East Pacific SST record.**

**C. Additional verifications of lagged influences on climate**

Two additional verifications of solar lagged explanations of an arctic and a tropical climatic records.

1) One verification is based on the Barents sea (BS) climatic indexes, that appears to be happen around 250 years before global temperature. This lead of the BS justify an analysis based on a Northern Sweden cave that helps to provide an experimental GT forecast. It support another independent GT forecast (see below).

[Figure]

a)

b)
Figure C1. Barents sea (BS) SST records over the last 12 Kyr. a) Map showing the BS in the eastern side of the Arctic ocean entrance (north of Norway Sweden and Finland); b) The Holocene SST reconstructed at BS compared with its model based on lagged TSI.

2) Another verification of the lagged climate response is developed with Madagascar and Magdalene  Islands hydroclimate records which lag TSI by around 3600 years. These records show an African monsoon onset several centuries earlier than the East Asia Summer monsoon (previously analyzed lags).

[Figure]

Figure C2. Isotopic oxygen anomalies (upper graph) from a cave in Madagascar and Mascarene Islands (Li et al., 2020) lags TSI record (lower graph) by around 3600 years.

**D. Explanations and additional verification of the GT 21ˢᵗ Century forecast**

The importance of my work (ESD-2021-84) for the past global climate (expressed in five modelling records) is very important. However, the consequences of this work for the future global temperatures (GT) are of paramount importance. In order to better support the GT decomposition and amplification, here we present an explanation of a kind of "resonance" between volcanic activity and the GT residual after to be considered the solar and lunar lagged influences.

Here, the amplification of 40% of the residual component of global temperature (GT minus the solar and tidal (NASA) component (GT res1b)] is explained. This amplification, could be considered as a resonant response to lagged volcanic activity. The component GT res1b and the lagged global volcanism (Sigl et al, 2005) over the last millennium were displayed in Figure D1. The global volcanic activity, after being adjusted linearly to GT values and lagged 80 yrs presents recurrences of about 180 years that can, after 4 cycles, generates a resonance over the GT res1b component.

[Figure]

**Figure D1. The residual global temperature (GT- solar-tidal) and lagged volcanic activity. The volcanic activity is lagged 80 yrs and has the recurrence of 178 yrs, with a consequent potential to generate a resonance for the GTres1b.**

Given the mentioned importance of the GT forecast, an independent verification of the GT forecast is also provided. This independent verification of the 21st century GT forecast is based on independent information coming from the isotopic content of calcites of a cave located in Northern Sweden.

In order to support the GT forecast developed in this work, an independent forecast was evaluated. This GT forecast is based on information coming from isotopic anomalies of a northern Sweden cave over the Holocene. Figure D2 shows this forecast and a comparison with the previous forecast.

[Figure]

**Figure D2. Another GT forecast and its comparison. a) The isotopic record from calcites of a Northern Sweden cave over the last 4000 years lagged 270 years and adjusted to the GT record (integrated in this work). b) A zoom of these records but with adjusted only with the last 300 years: and c) the previous graph b) with the superimposed graph of the GT record forecasted previously in this work**

**E. Analysis of CO2, solar activity and climate**

Finally, a long-term analysis of the CO2 records **are** presented. This analysis shows two important aspects of CO2, relative ocurrences and potential causes: a) over the last 50 Kyr, the usual occurrence of CO2 previous to climate variations, when volcanic activity is below normal, and the contrary when volcanic activity is above normal, **b) over the Holocene, and c) over the last 60years** where the solar lagged influences on **CO2 records that explains their** variations.

Based on two studies developed in the last decade, Viaggi 2021 and Brown 2014, an analysis of the occurrences of CO2 relative to Earth´s climate is developed. Figure E1 shows the relative ocurrences of CO2 and climate variations filtered around 9.7 and 9.8 Kyr oscillations (Viaggi et al., 2021). Fig E1a clearly show a predominant occurrence of Climate before those of CO2. However when volcanic activity, reconstructed by Brown et al. (2014) and shown in Fig E1b, is above normal the CO2 occurs before climate variations. These facts emphasize the thermal oceanic origin of CO2.

[Figure]

a)

[Figure]

b)

**Figure E1. Climate, CO2 and volcanism variations over the last 50 Kyr. a) The CO2 and climate variations filtered around 9.7 and 9.8 Kyr oscillations (Viaggi et al., 2021), clearly show a predominant occurrence of Climate before those of CO2; and b) global volcanic activity reconstructed by Brown et al. (2014).**

It is also explored the possibility that solar activity generates lagged influence on the atmospheric CO2 content. **Figure E2 and E3 displays CO2 and solar lagged variations over the Holocene and the last 60 years, respectively**.

[Figure]

**Figure E2. A comparison over the Holocene of CO2 (Royer et al., 2001) and solar lagged variations (SS16), with lags greater than 2000 years.**

[Figure]

**Figure E3. CO2 variations and its solar model over the last 60 years. CO2 changes are monthly registered at Mauna Loa Hawaii, by NOAA & SIO, and its model based on solar activity (SSN) is lagged half a year.**

The obtained results of CO2, depicted in Figures E1, E2 and E3, strongly suggest: with a relative lead/lag influenced by volcanic activity, with a very good match with solar lagged influences, around 2100 yrs later, and with a lagged solar influences on atmospheric CO2 changes measured in Mauna Loa Hawaii, respectively.

All these characteristics suggest that the causes of CO2 variations are of solar origin with different thermal and circulation mechanisms.

**REFERENCES**

Brown et al.: Characterisation of the Quaternary eruption record: analysis of the Large Magnitude Explosive **Volcanic** Eruptions (LaMEVE) database. Journal of Applied Volcanology 2014, 3:5

Clark, P.U., He, F., Golledge, N.R. et al. Oceanic forcing of penultimate deglacial and last interglacial sea-level rise. Nature 577, 660–664 (2020). https://doi.org/10.1038/s41586-020-1931-7

Fox, L. R., Wade, B. S., Holbourn, A., Leng, M. J., & Bhatia, R.: Temperature gradients across the Pacific Ocean during the middle Miocene. Paleoceanography and Paleoclimatology, 36, e2020PA003924. https://doi.org/10.1029/2020PA003924, 2021.

Lal, D., A.J.T. Jull, D. Pollard, and L. Vacher. 2005. Evidence for large century time-scale changes in solar activity in the past 32 Kyr, based on in-situ cosmogenic 14C in ice at Summit, Greenland. Earth and Planetary Science Letters 234, pp. 335-349.

Lea, D. W., Pak, D. K., Spero, H. J.: Climate impact of late quaternary equatorial Pacific sea surface temperature variations. Science 289, 1719–1724, 2000.

Lea D. W., et al., Paleoclimate history of Galápagos surface waters over the last 135,000 yr. Quat. Sci. Rev. 25, 1152–1167, 2006.

Li, Hanying, Sinha, Ashish, Anquetil André, Aurèle, Spötl, Christoph, Vonhof, Hubert B., Meunier, Arnaud, Kathayat, Gayatri, Duan, Pengzhen, Voarintsoa, Ny Riavo, Ning, Youfeng, Biswas, Jayant, Hu, Peng, Li, Xianglei, Sha, Lijuan, Zhao, Jingyao, Edwards, R. Lawrence, & Cheng, Hai.: A multimillennial climatic context for the megafaunal extinctions in Madagascar and Mascarene Islands. Science Advances, 6 (42). https://doi.org/10.1126/sciadv.abb2459, 2020

Risebrobakken, Bjørg; Dokken, Trond; Smedsrud, Lars Henrik; Andersson, Carin; Jansen, Eystein; Moros, Matthias; Ivanova, Elena V (2011): Early Holocene temperature variability in the Nordic Seas: The role of oceanic heat advection versus changes in orbital forcing. Paleoceanography, 26(4), PA4206, https://doi.org/10.1029/2011PA002117. DATA: Risebrobakken, Bjørg; Dokken, Trond; Smedsrud, Lars Henrik; Andersson Dahl, Carin; Jansen, Eystein; Moros, Matthias; Ivanova, Elena V; Sarnthein, Michael (2019): Calculated foraminiferal temperatures, and relative abundance, sediment core GIK 23258-2. PANGAEA, https://doi.org/10.1594/PANGAEA.904941,

Royer D.L., et al. 2001, **CO2** variations  Earth Science Reviews 54, pp 349-392

Sundqvist, H. S., Holmgren, K., Moberg, A., Spö¨ tl, C. & Mangini, A.: Stable isotopes in a stalagmite from NW Sweden document environmental changes over the past 4000 years. Boreas, Vol. 39, pp. 77–86. doi:10.1111/j.1502-3885. 2009.00099, 2010

Tierney, J. E., N. J. Abram, K. J. Anchukaitis, M. N. Evans, C.Giry, K. H. Kilbourne, C. P. Saenger, H. C. Wu, and J. Zinke (2015), Tropical sea surface temperatures for the past four centuries reconstructed from coral archives, Paleoceanography, 30, 226–252, doi:10.1002/2014PA002717.

Ueno, Y., Hyodo, M., Yang, T. et al. Intensified East Asian winter monsoon during the last geomagnetic reversal transition. Sci Rep 9, 9389 (2019). https://doi.org/10.1038/s41598-019-45466-8

Viaggi P. (2021) Quantitative impact of astronomical and sun-related cycles on the Pleistocene climate system from Antarctica records. Quaternary Science Advances, Volume 4, October 2021. Supplementary data set: https://ars.els-cdn.com/content/image/1-s2.0-S2666033421000162-mmc1.xlsx [Original data from: Past Interglacials Working Group of PAGES (2016), Interglacials of the last 800,000 years. Rev. Geophys. 54:162–219, https://doi.org/10.1002/2015RG000482]

---

## Author Comment (AC2)

**Comment 2: Modelling and forecast of global climate (sea-level [SL] and ocean-heat-content [OHC]) based on mutimillennia lagged and recurrent solar influences.**

In order to continue complementing the *ESD-2021-84* paper in discussion, the present comment 2 is associated with two of the main global climate variables, sea-level [SL] and ocean-heat-content [OHC, estimated with the ocean mean temparature (MOT)].

These two global climate variables are explained and forecasted taking into account the mutimillennia lagged and recurrent solar influences. As the five variables firstly analyzed in the *ESD-2021-84* paper, these two oceanic variables, are also explained with the slow oceanic transports mentioned in discussion, however these two variables require both longer accumulated distances and their associated "travel times", considering the surface and subsurface flows of the well known OCB.

Then these two mentioned global climate variables will be analyzed. Firstly, sea level records have been reconstructed using several techniques. One important technique considered the Red Sea as a reservoir that "register" SL variations for the Holocene (Sidall et al., 2003; S03). Another, independent reconstruction of SL for the last two and a half millennia is considered (Kopp, et al., 2016; K16)

The model employed to express the SL variability is the model 1, shown in the main document of this work (ESD-2021-84). Using this model, SL is expressed as a lagged linear transformation and trend adjusted of solar activity, the lag applied is 5.7 Kyr. This is shown in Figure 2.1. Although there are periods, for instance 8.6-6.6 Kyr BP, with significant differences, the rest of the Holocene SL is explained well with the adjusted model.  This model also provided information to forecast SL for the next two millennia with oscillating but increasing trend.

[Figure]

**Figure 2.1  Modelling and forecast of SL record over the last 12 and future 2 Kyr, respectively. The SL record (S93) is modelled based on the TSI record (SS16).**

Anoher model is employed to express the SL variability but considering Greenland temperatures (GrT). Using also model 1, SL is expressed as a lagged linear transformation and trend adjusted to GrT reconstructed record (Kobashi, 2018) the lag applied is 1.8 Kyr. This is shown in Figure 2.2 based on a lag for GrT of around 1.8 Kyr. The lag respect the solar activity is evaluated adding 3.85 Kyr that is the estimated lag (See Table1 of the *ESD-2021-84* paper). Then the lag respect TSI is estimated as (3.85+1.8 Kyr) 5.65 Kyr.

This model also provided information to forecast SL for the next two millennia with oscillating and increasing trend only in the first next centuries, and following with a decreasing trend.

[Figure]

**Figure 2.2 Modelling and forecast of SL record over the last 8 and future 2 Kyr. The SL record (S03) is modelled based on the GrT record (K18).**

The two SL models employed show two catastrophic scenarios for the SL variability for the next centuries. However in these two graphs, 2.1 and 2.2, the SL record developed by S03 shows a decreasing trend over the Holocene. An important comparison is shown in Figure 2.3 between the SL S03 record and the second model SL(GrT), for the last 2.5 Kyr, with a more recent SL reconstruction provided by K16.

This comparison, of SL records show two important aspects. First, one is a huge decreasing of sea level changes respect those presented some millenia before. For instance, the values estimated with the SL K16 reconstruction, are around 100 times lower than those estimated by the SL S03 [meters to centimeters]. And second, there is a the good modelling performance of the SL(GrT) for the more recent SL reconstruction developed by K16.

[Figure]

**Figure 2.3 Comparison of SL records and model for the last 2.5 Kyr. SL S03 (Dark blue line) reconstruction [with small numbers in Kyr BP and meters ] and its model SL(GrT) are compared with SL K16 (Gray line) [with greater numbers in Calendar Year and SL in centimeters]. The SL(GrT) model follow the Kopp values of SL that indicate SL values 100 times lower than those estimated by Sidall.**

The second variable to be analyzed is the ocean-heat-content (OHC) or mean ocean temperature (MOT). There are two MOT records that have been reconstructed using a technique based on the rare gases content in ice-cores. These gases "registered" MOT variations both for the Eemian (Shackleton, et al., 2019; S19) and the Holocene (Bereiter et al., 2015; B15) periods.

The model employed to express the OHC variability is, also the model 1, shown in the main document of this work (ESD-2021-84). Using this model, MOT is expressed as a lagged linear transformation and trend adjusted to solar activity. On one side, Figure 2.4 shows the analysis for the Eemian MOT considering a lag of around 7.5 Kyr.

[Figure]

a)

[Figure]

b)

**Figure 2.4  Graphical modelling and forecast of OHC (or MOT) record over the 155 to 120 [Ky BP] based on recurrent and lagged TSI variation (matching TSI time at 0 yr BP with 116Kyr BP [9.5 Kyr*13-7.5 Kyr] of Eemian reconstructed record S19); a) the TSI record matching the Eemian interglacial; b)  the TSI record matching the 20 Kyr period prevoius to the Eemian.**

On the other side, Figure 2.5 shows the analysis of deglaciation and Holocene MOT record, considering a lag of around 7 Kyr.

[Figure]

**Figure 2.5  Modelling and forecast of MOT from WAIS ice-core (B18) and its modelling based on lagged TSI variation. Green squares indicate Ocean heat content converted to MOT values  for the Pacific Ocean (Rosenthal et al., 2013).The TSI lag employed is 7.0 Kyr.**

The delayed solar influences on both SL and MOT variations provide additional elements to better model climate variabilities, and more than 2000 years of global marine climate forecast.

The two presented global forecasts, for SL and MOT, provide further elements to accept the OCB with its surface and intermediate ocean circulations where thermal anomalies are transported, to all the world.The model for lags of solar influences and their accumulated distance,complemented with two red points corresponding to SL and MOT is shown in Figure 2.6. The sea level with a lag of 5.7 Kyr,is associated with an accumulated distance of 16 Earth's Equatorial Circumference (EEC)/4 when the deep flows go around the Antarctic continent. The mean ocean temperature with a lag of 7.2 Kyr, is associated with an accumulated distance of 19 of EEC/4 when the deep flows go around the North Pacific. [Please note that EP is changed by EEC]

[Figure]

**Figure 2.6 Modelling of OCB-lagged climate connections and its graphical surface and deep ocean currents. a) the lags and accumulated distance model for surface and deep flows, including with red circles the SL and MOT lags and their accumlated distances; b) the graphical representation of the OCB with surface (light blue) and deep (dark blue) circulations.**

REFERENCES

Bereiter, B., Shackleton, S., Baggenstos, D., Kawamura, K. & Severinghaus, J. Mean global ocean temperatures during the last glacial transition. Nature 553, 39–44, 2018.

Kopp R.E., Kemp A.C., Bittermann K., Horton B.P., Donnelly J.P., Gehrels W.R., Hay C.C., Mitrovica J.X., Morrow E.D., and Rahmstorf S., Temperature-driven global sea-level variability in the Common Era, Proceedings of the National Academy of Sciences, vol. 113, pp. E1434-E1441, http://dx.doi.org/10.1073/pnas.1517056113, 2016

Rosenthal Y., Linsley B.K. and Oppo D.W.,Pacific Ocean Heat Content During the Past 10,000 Years. Science 342(6158), pp. 617-621, doi:10.1126/science.1240837, 2013.

Sánchez-Sesma, J.: Evidence of cosmic recurrent and lagged millennia-scale patterns and consequent forecasts: multi-scale responses of solar activity (SA) to planetary gravitational forcing (PGF), Earth Syst. Dynam., 7, 583-595, 2016.

Shackleton, S., Baggenstos, D., Menking, J., Dyonisius, M., Bereiter, B., Bauska, T., Rhodes, R., et al.: Global ocean heat content in the Last Interglacial. Nature Geoscience, 13 (1), 77-81. https://doi.org/10.1038/s41561-019-0498-0, 2020.

Siddall M, Rohling EJ, Almogi-Labin A, Hemleben Ch, Meischner D, Schmelzer I & Smeed DA, Sea-level fluctuations during the last glacial cycle. Nature, 423(19) , 2003

---

## Author Comment (AC3)

**Comment 3: Model and forecast of global climate (Rain and temperature) based on multidecada lagged responses to SST-epac changes.**

In order to complement the ESD-2021-84 paper in discussion with shorter temporal scales, the present comment is associated to the global and monsoon rainfalls and its lagged influences of Eastern Pacific (EPac).

Although the atmosphere-ocean connection between the EEPac and the hydroclimate of the rest of the world is not as slow as the oceanic transports mentioned in the ESD-2021-84 paper in discussion, however the "trade winds" and the associated surface currents, transport the heat and water vapor to the western regions in a "faster way" that influence rainfalls several decades later.

The three largest continents, Asia, America and Africa, are affected by monsoon rains (Wang et al., 2012; 2020). As part of the ESD-2021-84 work, delayed solar influence are analyzed on rainfall for four countries, located in those continents, and for global scales. The annual accumulated rainfalls, coming from CRU/UEA(2022), in four countries: India, China, Nigeria and Mexico (IN, CH, NG & MX) over the period 1901-2020 are analyzed. The rains mentioned are shown in Figure 3.1.

[Figure]

**Figure 3.1  Annual accumulated rainfalls, coming from CRU/UEA(2022), in four countries: India, China, Nigeria and Mexico (PR-IN, PR-CH, PR-NG & PR-MX) over the period 1901-2020.**

The annual accumulated rainfalls in IN, CH, NG and MX, are analyzed with respect to the SSTepac, record. Figures 3.2-3.5 shows the explanation of four national accumulated rainfalls based on solar influence (expressed through the SSTepac). The delays of those national rainfall records of India (IN), China (CH), Nigeria (NG) and Mexico (MX), respect the SSTepac, are 40, 35, 51 and 41 years, respectively.

[Figure]

**Figure 3.2  Annual accumulated rainfalls, coming from CRU/UEA(2022) for India (PR-IN) over the period 1901-2020 and extended thanks to the SSTepac records, and its lagged influences.**

[Figure]

**Figure 3.3  Annual accumulated rainfalls, coming from CRU/UEA(2022) for China (PR-CH) over the period 1901-2020 and extended thanks to the SSTepac records, and its lagged influences.**

[Figure]

**Figure 3.4 Annual accumulated rainfalls, coming from CRU/UEA(2022) for Nigeria (PR-NG) over the period 1901-2020 and extended thanks to the SSTepac records, and its lagged influences.**

[Figure]

**Figure 3.5 Annual accumulated rainfalls, coming from CRU/UEA(2022) for Mexico (PR-MX) over the period 1901-2020 and extended thanks to the SSTepac records, and its lagged influences.**

The importance of delayed SSTepac influences shoud be evaluated not only in the four analyzed country records associated with the global monsoon, but in the global rainfall.

To better develop this global anaylsis it is convenient to extent the SSTepac record considering the ENSO EN12 SST record. It is displayed in Figure 3.6 where SSTepac values are extrapolated, thanks to N12 SST values, after to be calibrated over the period 1856-2000.

[Figure]

**Figure 3.6  SSTepac compared with Niño12 SST. The annual mean N12-SST values help to extend the SSTepac record over the period 2000-2021.**

We have employed the KNMI Climate Explorer (KNMI, 2022) provides a Spatial statistic of NOAA/NCDC gridded GHCN v2 for global land precipitation anomalies. The global average values were estimated through the same database (KNMI, 2022) but using the CRU/UEA precipitation data over land. In Figure 3.7, the global precipitation PG, is expressed as a linear function of the lagged eastern Pacific sea surface temperature [SSTepac]. The lag employed is 43 years, that minimize the error between PrecGLO and its model.

[Figure]

**Figure 3.7  Annual accumulated global rainfalls, coming from WMO(2022) over the period 1901-2020 and extended thanks to the SSTepac record (this record was also extended based on N12 record) and its lagged influences.**

The extended  SSTepac record provides more than 40 years of global rainfall forecast.

In Figure 3.8, the global temperature (GT), is expressed as a linear function of the lagged eastern Pacific sea surface temperature [SSTepac]. The lag employed is 23 years, that minimize the error between TG and its model.

[Figure]

**Figure 3.8   Global temperature coming from HadCRUT5 (2022) over the period 1856-2021, is modelled and forecast thanks to the extended SSTepac record (See Figure 3.6). The lag imposed to the model [GT(SSTepac)] is 23 years lagged influences.**

The extended  SSTepac record provides more than 20 years of global temperature forecast.

The two presented global forecasts, for rainfall and temperature, provide further elements to accept ocean and atmosphere circulations where the SST anomalies are transported, toward the West, to all the rest of the world.

As a final comment, considering the numerous lagged responses analyzed, I would like to expose a capacity of the IPCC scenarios that has not been considered. I present the linearly extrapolated and averaged version of the rcp2.6, rcp4.5 and rcp8.5 scenarios to estimate the rcp0.0 scenario.

Using this two simple formulas:

rcp0.0a(t)=rcp2.6(t)-2.6*(rcp4.5(t)-rcp2.6(t))/(4.5-2.6)
rcp0.0b(t)=rcp2.6(t)-2.6*(rcp8.5(t)-rcp2.6(t))/(8.5-2.6)

and after averaging these each pair of values (1850-2100), is posible to estimate an approximated official scenario for rcp0.0.

This scenario, that shows a modelling toward conditions associated with "zero GHG emissions", is shown in Figure 3.9 and compared with the initial estimated scenario from Figure 11b.

[Figure]

**Figure 3.9   Global temperature scenarios 1850-2100. An adapted Figure 11b from the paper ESD-2021-84 in discussion with the Average rcp0.0 values (in yellow line).**

REFERENCES

CRU/UEA data: a) rainfall by country. https://data.ceda.ac.uk/badc/cru/data/cru_cy/cru_cy-_4.05/; retrived 01 February 2022; b) global temperature.

KNMI, 2022, The Koninklijk Nederlands Meteorologisch Instituut (Royal Netherlands Meteorological Institute), international climate database. https://climexp.knmi.nl/get_index.cgi. Retrived 01 February 2022

PAGES, 2022: Global Monsoon Working Group. Webpage:https://pastglobalchanges.org/science/wg/former/globalmonsoon/intro, retrived 03 February 2023.

Tierney, J. E., N. J. Abram, K. J. Anchukaitis, M. N. Evans, C.Giry, K. H. Kilbourne, C. P. Saenger, H. C. Wu, and J. Zinke (2015), Tropical sea surface temperatures for the past four centuries reconstructed from coral archives, Paleoceanography, 30, 226–252, doi:10.1002/2014PA002717.

Wang, P. X., B. Wang, H. Cheng, J. Fasullo, Z. Guo, T. Kiefer, and Z. Liu, 2017: The global monsoon across time scales: Mechanisms and outstanding issues. Earth-Sci. Rev., 174, 84–121, https://doi.org/10.1016/j.earscirev.2017.07.006.

Wang B., Jin C., and Liu J., Understanding Future Change of Global Monsoons Projected by CMIP6 Models, J. Clim.  33 (15) p6471–6489, doi: https://doi.org/10.1175/JCLI-D-19-0993.1, 2020.

---

## Author Comment (AC4)

**Comment 4: Model and forecast of Arctic Summer and Central England temperatures and their use for empirical 21st century global temperature forecasts**

In order to continue complementing the ESD-2021-84 paper in discussion, with shorter temporal scales, the present comment is mainly associated with the Arctic and central England climate modelling and their influences on global climate.

**Part 1. Arctic Summer Temperature (AST)**

DATA. The Arctic summer temperature (AST or T) information is based on a recently reconstructed record obtained from several proxies (ice-core, lake sediments and tree-ring), from 23 different local and regional processes (Kaufman et al., 2009). This record and another auxiliar record from Greenland temperatures (Vinther et al., 2009) are depicted in Figure 4.1.1.

**Figure 4.1.1** Millennia-scale reconstructed Arctic and Greenland Temperatures. a) Reconstructed Arctic Summer Temperatures (AST) anomalies [°C], obtained from 23 different proxies for the last 2,000 years (Kaufman et al., 2009), and b) Greenland temperature (GrT) during the late Holocene (Vinther et al., 2009).

MODELS. Tacking into account the recurrent and complex climate environment the AST, or T, is modelled with Fourier Series (FS) and a symetric lagged models. To take into account different contributions, the smoothed climatic variable T may be decomposed into two components of linear and non-linear oscillations, and their calculated error as follows:

$$T(t) = T_{FS}(t) + T_{NL}(t) + e(t) , \qquad (1)$$

where  $T_{FS}(t)$  is the Fourier series (FS) component due to multi-centennial fluctuations,  $T_{NL}(t)$  is the non-linear component based on symmetric self-similarity, and e(t) is the error. The FS component may be written by means of:

$$T_{FS}(t) = \sum_{j=1}^{N_{FS}} \left[ a_j \cdot \sin\left(j\frac{2\pi(t)}{P}\right) \right] + \sum_{j=1}^{N_{FS}} \left[ b_j \cdot \cos\left(j\frac{2\pi(t)}{P}\right) \right] + c, \qquad (2)$$

Here, *P* is the FS basic period,  $N_{FS}$  represents the number of FS terms or harmonics, j is an index component term, *a*, *b* and *c* are amplitude and bias parameters, and *t* is time.

In addition, to take into account recurrent patterns with symmetric/negative contributions, the *T* non-linear variation may be modeled with the following lagged and symmetric expression:

$$T_{NL}(t) = -\alpha T_{NL}(t - \delta_t) + \delta_T, \qquad (3)$$

with  $\alpha > 0$ ;

where  $\alpha$  is the symmetric amplification factor,  $\delta$  is the temporal lag, or symmetric period, and  $\delta_T$  is the bias of this component.

Assuming  $N_{FS}$ , the constants *a*, *b* and *c*, and period *P* may be jointly evaluated after a multi-linear regression that looks for a basic FS period that minimizes the RMS values of e(t). The second component of the *T* signal is evaluated, based on the symmetric self-similarity of this component and another minimization of errors.

RESULTS. To estimate climate oscillations in the reconstructed record *T*, or smoothed AST, firstly a wavelet analysis was applied using the online resource by Torrence and Compo (1998). The spectral analysis results of the AST, based on wavelets and displayed in Figure 4.1.2, show three main and significant (10% significance level) periodicities around 2000, 1000 and 500 years. Two secondary oscillatory contributions were also detected at periods around 180 and 63 years.

---

## Author Comment (AC5)

**Comment 5: Greenland and Global reconstructed temperatures and their use for empirical 21$^{st}$ century global temperature forecasts**

In order to provide more complementary information to the ESD-2021-84 paper in discussion, with shorter temporal scales, the present comment is associated with the Greenland and Global temperature and their influences on global climate.

**Part 1. Greenland Temperature (GrT)**

DATA. The Greenland temperature (GrT) information is based on ice-core analysis,  (Kobashi et al., 2017).

RESULTS. To estimate global climate oscillations in the reconstructed record GrT, a linear and lagged transformation is adjusted to GT record. The GT model based on GrT that was linearly adjusted considering a lag of 75 years, is shown in in Figure 5.1.1. An additional bias for values later 1750 AD was applied, it is shown without and with bias application in parts a) and b) of Figure 5.1.1, respectively.

[Figure]

a)

b)

**Figure 5.1.1 Modelling of GT based on Greenland Temperature reconstructed record (K18). Linear adjusted and lagged (75 yr) GrT record a) withouth, and b) with bias correction (+0.55 °C) over the last 250 years.**

It is important to mention that the use of constant bias to adjust GrT information to GT records over more than the last two centuries emphsize two aspcts:

1) a cooling event around Greenland that influence its temperatures, and
2) a constant connection between GrT and GT over the last 250 years.

The coresponding 21st century scenario provided by the transformed, lagged and adjusted GrT record is  shown in Figure 5.1.2

[Figure]

**Figure 5.1.2  Global temperatures and its empirical forecast 2010-2100. A linearly adjusted and lagged GrT annual values to the GT integrated record 238-2010; a forecast for the next five decades is shown.**

**Part 2. Global temperature.**

DATA. The Global temperature (GT) information was integrated in main part of ESD-2021-84 paper.

RESULTS. To estimate global climate oscillations in the reconstructed record GT, an analog modelling with linear and lagged (around 1300 yrs later) are applied. Its results, with an amplification of  around 5 times and its simple verification, are shown in Figure 5.2.1

[Figure]

a)

b)

**Figure 5.2.1  Global temperatures and its empirical forecast 2010-3000. a) A linearly adjusted and lagged GrT annual values to the GT integrated record 238-2010; a forecast for the next millennium is shown; b) In order to verify, the GT(GrT) shown in a), is overlapped on the crbT record image (with times in Kyr BP), wich is almost directly affected by solar activity. Then its possible to verify lagged influences on GT**

**more than 4000 years later that are also depicted and show a long-term cooling increasing-trend for the next centuries.**

[Figure]

**Figure 5.2.2  Global temperatures and its empirical forecast 2020-2100. A linearly adjusted and lagged model, of the same GT integrated record 238-2010; a forecast for the next decades is shown.**

**Part 3. New Global temperature scenarios.**

The next table shows a comparison of all scenarios estimated in this study (main part of ESD-2021-84 paper in discusion and its comments).

| Scenario | Modelling elements | See details In |
|---|---|---|
| 0 | solar, lunar and other shorter recurrent contributions | main part of ESD-2021-84 paper |
| 1 | calcites of a Northern Sweden cave over the last 4000 years lagged 270 years [with a bias over the last centuries] | comment 1 of the ESD-2021-84 paper |
| 2 | SSTepac (extended with Niño 12 recent data) | comment 2 of the ESD-2021-84 paper |
| 3 | IPCC scenarios (an average of extrapolated version of the rcp2.6, rcp4.5 and rcp8.5 scenarios to estimate the rcp0.0 scenario. This scenario correspond to  "zero GHG emissions" at the end of the 21$^{st}$ century. ) | comment 3 of the ESD-2021-84 paper |
| 4 | Arctic temperature (AT) | comment 4 of the ESD-2021-84 paper |
| 5 | Central England Temperature (CET) | comment 4 of the ESD-2021-84 paper |
| 6 | Greenland temperature [with a bias over the last 250 yrs] | This comment |
| 7 | Global tempertaure (integrated record) | This comment |

A simple comparison of the last two scenarios presented in this comment is shown in Figure 5.3.1. The two new scenarios (6 and 7) present, similarly as scenarios 1, 2, 3 and 4, a decaying values for the next decades.

[Figure]

**a)**

**b)**

**c)**

**Figure 5.3.1  Comparison of global temperature scenarios 1850-2100. An adapted Figure 11b from the paper ESD-2021-84 in discussion with the first scenario estimated (Scenario 0, with cyan line), the Average rcp0.0 values (Scenario 3 in yellow line) and Arctic temperatures (T) (Scenario 4, in red wide line), with the ajusted and lagged forecast based on: a) Greenland temparature (Scenario 6, in ligth blue line); b) the Global temperature (GT) (Scenario 7, in red thin line).**

**Part 4. Preliminary conclusions (of this and previous coments)**

Our results, not only are providing general cooling trend scenarios for the rest of the 21$^{st}$ century, but also support that, non-linearity and multi-scale modeling efforts are required for accurate modeling and forecasting of climate-related issues.

Our results also have placed astronomical low-frequency (with multi-millenial and multi-centennial) processes, with its long-term and delayed influences (with ocean-atmospheric mechanisms) on climate variability in an important place for climate modeling and analysis, and of course, for climate forecasting.

REFERENCES

Kobashi, T., Menviel, L., Jeltsch-Thömmes, A. et al. Volcanic influence on centennial to millennial Holocene Greenland temperature change. Sci Rep 7, 1441. https://doi.org/10.1038/s41598-017-01451-7, 2017

---

## Author Comment (AC6)

**Comment 6: Model and forecast of Central England temperatures and Atlantic tropical cyclone intensities, and their use for empirical 21$^{st}$ century global temperature forecasts**

In order to continue complementing the ESD-2021-84 paper in discussion, with shorter temporal scales, the present comment is mainly associated with the central England and Atlantic tropical cyclone climate reconstructions and their influences on global climate.

**Part 1. Central England Temperature (CET)**

DATA. The Central England temperature (CET) series is the longest instrumental time series of temperature in the world (Parker et al 1992), with monthly values that extends back to 1659 (MetOffice, 2022). CET annual anomaly values are depicted in Figure 6.2.1.

MODEL. Although during  last decades there has been a rapid warming in the CET that appears to be an anthropogenic influence on the climate, however here a new and non-linear Atlantic Multidecadal Oscillation (AMO) model is proposed and applied for the CET.

RESULTS. The modelling non-linear results of the CET are also displayed in Figure 6.2.1 and provide an empirical forecast for annual CET values. The CET non-linear model based on the AMO reconstructed record (Gray et al., 2004) is adjusted linearly and lagged 80 years.

[Figure]

**Figure 6.1.1  Central England temperatures (CET) 1660-2100 in annual values. The observed anomalies are modelled considering a AMO lagged analog located  80 yrs after. The analog model is adjusted with a linear transformation, a forced slope and a temporal lineal reduction adjustment of 5%.**

APLICATION. The modelling non-linear results of the CET are adjusted to the global temperature (GT) HadCRUT5 record. The GT instrumental  recordand the GT(CET) extrapolated record are also displayed in Figure 6.1.2  an empirical forecast for annual GT values.

[Figure]

**Figure 6.1.2  Global temperatures and its empirical forecast 2010-2100. a) Non-linearly adjusted CET record in annual values to the GT integrated record 238-2010, a temporal expansion of 15% was applied; b) a zoom of a) for the last 5 and next centuries.**

This GT[CET(AMO)] model with a lag of around 80 years, can be justified considering previous results. The OCB in its surface currents affects firstly the NW Atlantic and later the NE Atlantic.

**Part 2. Tropical cyclone intensification Index (TC-II)**

In the following the main aspects of the extended abstract: "An Atlantic Tropical Cyclone Intensification Index for the last 2000 yr: A Significant ~510 yr Climate Cyclic Pulse Reconstructed" presented by the author in the AMS meeting at Orlando, Florida (Sánchez-Sesma, 2008, SS08 hereafter).

Based on historical and instrumental Atlantic Tropical Cyclones (ATC) recently updated records, and hydrological proxy records from the Caribbean, this paper proposes to reconstruct an ATC intensification index (II) for the last 2 millennia

DATA AND METHODS. Three main sources of ATC information were used in this work, one is from the instrumental measurements covering the last one and a half centuries [HURDAT (Landsea, 2005), HD hereafter], a second is the reconstructed record recently updated with Spanish documentary sources for the 1500-1850 period [García-Herrera et al. (2005), GH hereafter], and a third is the oxygen-isotope monsoon rainfall record from U/Thdated stalagmite from the Isthmus of Panama (Lachniet et al, 2004, LA07 hereafter). Instrumental, historical and proxy raw information is displayed in Fig. 6.2.1.

[Figure]

Figure 6.2.1. Raw observed and reconstructed information of TC & storms. a) Annual accumulated number of Atlantic Tropical Cyclones (ATC) as recorded by NOAA in the HURDAT data base (Landsea, 2005); b) Annual accumulated number of Atlantic TCs (ATC) from documentary sources for the period 1500-1850. (García-Herrera et al., 2005); c) A Raw δ18O values of speleothem calcite from Chilibrillo, Panama (LA04).

*An Intensification Index(II).* Although there is a possible bias due to missing ATC information, its reconstruction during the last 500 years provides important information. If we analyze the relative values of the hurricanes (HR) and tropical storms (TS) collected by GH and HD we can estimate trends and oscillations. To do that, an intensification index for TC is proposed as the following (SS08):

$$II(t)=[HR(t)-TS(t)]/ [HR(t)+TS(t)] \qquad (6.1)$$

where II(t) is the annual intensification index value,
HR(t) is the annual number of hurricanes in the ATC records, TS(t) is the annual number of tropical storms in the ATC records, and t is the corresponding year.

The model limits II values which can oscillate between -1.0 and 1.0, when there are only tropical storms or hurricanes, respectively. II values do not change if the same factor is applied both in HR and TS. Thiis aspect is very important to cope with limitations of the colonial historical observations where subestimations could be present due to limited coverage of the Spanish float vayages and land reports.

*Reconstruction of II for the last millennia.* We evaluated the ATC II defined in eq. 1 over the period as 1500-2006. We applied a moving average filter (MA-21-yr) to the ATC II values to emphasize multidecadal processes. Employing GH and HURDAT archives of historical and instrumental information, respectively, it is possible to evaluate the almost complete ATC II history of the last 500 years. The reconstructed ATC-II values are shown in Figure 6.2.2a. An iterative process that looks for the minimum RMS error adjusted to ATC-II values a sine function with a period of 496 years; it is also shown in this Figure. 6.2.2a. The Atlantic Ocean intensification index (ATC II) defined in Equation 6.1 with annual accumulated values filtered with a 41-yr moving average are displayed in Figue 6.2.1a. In the same Figure, an adjusted sine function with a 496 yr period is also displayed.

[Figure]

**Figure 6.2.2. Results for the Atlantic TC intensification index (ATC II). a) ATC-II based on instrumental and historical information; b) ATC-II based on instrumental, historical and detrended proxy information. In both Figures FS models with P=496 yrs,Ns=1 and P=508yrs, Ns=2, are also displayed in a and b) respectively. Model 0 was based on all data from 231 BC to 1985 A.D. In contrast, model 1 began in the same initial year, 231 BC, but ended in 1310 A.D. The models 0 and 1, explained 67.7 and 65.4% of variance for all the reconstructed period.**

Figure 6.2.2b displays ATC-II linearly transformed and 81-yr moving-average proxy record for the speleothem based Central America monsoon record over 1500 yr (180 B.C.-1310 A.D.) obtained by LA04, with the historical and instrumental reconstruction, based on GH and HD data, using a 41-yr moving average on the ATC-II data as previously made.

In order to evaluate the stability of the 500 yr oscillation Fourier series with N=2 was applied both to the detrended Caribbean proxy isotope-based (LA04) and to the directly evaluated ATC-II record. Also, in Figure 6.2.2b two adjusted Fourier series models with period of 508 yrs are displayed. Model 0 was based on all data from 231 BC to 1985 A.D. In contrast, model 1 began in the same initial year, 231 BC, but ended in 1310 A.D. The models 0 and 1, explained 67.7 and 65.4% of variance for all the reconstructed period.

[Figure]

Figure 6.2.3. Surface streamlines for the May-October season during the past 51 years (1955-2005 period) based on the reanalysis information (Kalnay et al 1996).

The importance of the Panamanian stalagmite record for ATC not only is shown with its agreement with tha historical and instrumental ATC-II records, but also is shown, when the mean streamlines at 1000 mb level are displayed for the Caribbean. Based on the reanalysis information (Kalnay et al 1996), Figure 6.2.3 displays these streamlines for the May-October season during the 1955-2005 period. This graphical result shows the influence of the Caribbean (or the main development region of ATC) in the Panamanian zone in which a semi-permanent low center is located.

APLICATION. The integrated record of the ATC-II are adjusted to the global temperature (GT) reconstructed record (Singh et al., 2018). The lagged influences on GT of the ATC-II can be appreciated in the Figure 6.2.4 where the GT reconstructed record (Singh et al., 2018) is overpossed on the lagged (150 yr) ATC-II reconstructed record. This empirical model confirm a GT cooling trend for the next decades.

[Figure]

**Figure 6.2.4. Global temperatures (Singh et al., 2018) and its empirical forecast 2010-2150. The adjusted model is based on reconstructed and integrated ATC-II.**

**Part 3. New global temperature scenarios based on CET and ATC-II.**

A simple comparison of new scenarios estimated in this study (main part of ESD-2021-84 paper in discusion and its four comments) are shown in Figure 6.3.1

The new GT scenarios, are evaluated considering CET(AMO) and ATC-II as was explained in this comment before.

[Figure]

[Figure]

**c)**

**Figure 6.3.1   Comparison of six global temperature scenarios 1850-2100. An adapted Figure 11b from the paper ESD-2021-84 in discussion with the first scenario estimated (Scenario 0, with cyan line) and the Average rcp0.0 values (Scenario 3 in yellow line) and the modelling of Arctic temperatures (T) (Scenario 4, in red wide line); b) the modelling of central England temperatures (CET[AMO]) (Scenario 7, in red dotted line); c) [a zoom of the Figure 6.2.4] the modelling of ATC-II (Scenario 8, in black line)**

**Part 4. Preliminary additional conclusions**

The two new GT scenarios not only have shown decreasing tempertaures for the next decades,but also confirm the OCB influences over the Atlantic realm.

The two GT models, based on ATC-II and CET(AMO), shown different lags of 150 and 80 years, respectively, that can be justified considering previous results of the OCB influences due to its surface currents. The OCB firstly influences the tropical Atlantic and later the NW Atlantic, and later influence the North Atlantic and global temperature.

ACKNOWLEDGMENTS (2008):
The author would like to express his gratitude to Dr. Craig Loehl, Dr. William Perry, Dr. Frederic Vitart, and Dr. Eduardo Zorita their important comments and suggestions, and also to Dr. Ricardo García-Herrera, and Dr. Matthew Lachniet their valuable information for the development of this work. This work was carried out with the aid of grants from the Inter-American Institute for Global Change Research (IAI) 03SGP211-214 and CRN-II-#2050, which are supported by the US National Science Foundation (Grants GEO- 0341783 and GEO-0452325, respectively).

REFERENCES

*CET&AMO*

Gray, S., Graumlich, L., Betancourt, J., and Pederson, G.: A tree-ring based reconsruction of the Atlantic Multidecadal Oscillation since 1567 AD, Geophys. Res. Lett., 31, L12205, https://doi.org/10.1029/2004GL019932, 2004.

Met Office U.K. "Monthly Mean Central England Temperature (Degrees C)". Met Office, Hadley Centre for Climate Prediction and Research. Retrieved 11 March, 2022.

Parker, D. E., T. P. Legg, and C. K. Folland: A new daily Central England Temperature Series, 1772-1991. Int J Climatol, 12, 317-342, 1992.

Singh, H. K. A., Hakim, G. J., Tardif, R., Emile-Geay, J., and Noone, D. C.: Insights into Atlantic multidecadal variability using the Last Millennium Reanalysis framework, Clim. Past, 14, 157–174, https://doi.org/10.5194/cp-14-157-2018, 2018.

*TC-II*

García-Herrera R., L. Gimeno, P. Ribera, E. Hernández, 2005: New records of Atlantic hurricanes from Spanish documentary sources. J. Geophys. Res., 110, D03109, doi:10.1029/2004JD005272.

Lachniet, M.S., S.J. Burns, D.R. Piperno, Y. Asmerom, V.J. Polyak, C.M. Moy, and K. Christenson, 2004: A 1500-year El Niño/Southern Oscillation and rainfall history for the Isthmus of Panama from speleothem calcite. Journal of Geophysical Research, Atmospheres, 109, D20117, doi:10.1029/2004JD004694.

Landsea, C. W., C. Anderson, N. Charles, G. Clark, J. Dunion, J. Partagas, P. Hungerford, C Neumann and M. Zimmer, 2004: The Atlantic hurricane database reanalysis project: Documentation for the 1851-1910 alterations and additions to the HURDAT database. In Hurricanes and Typhoons: Past, Present and Future, R. J. Murnane and K.-B. Liu, Eds., Columbia University Press.

Liu, K-b., J. Sánchez-Sesma, J. P. Donnelly, A. L. Cohen, C.I. Mora, A. Frappier, E.J. Alfaro, J. Amador, and M. Peros, 2005: Paleotempestology of the Caribbean Region: A Multi-proxy, Multi-site Study of the Spatial and Temporal Variability of Caribbean Hurricane.Proposal of a research project supported by the Inter-American Institute for global change research (IAI) (Undercurrent CRN-II-#2050 project, budget $620,000 USD, period 2006-2010).

Sánchez-Sesma, J., et al. 2005: Evaluation of Paleo-Hurricanes in the Intra-Americas Sea: A reconstruction and Analysis Based on Proxy Records. Final Technical Report, Project 03SGP211-214, Interamerican Institute for Climate Change Research, IAI, 83p.

Sánchez-Sesma, J. et al, 2005: Evaluation of Paleo-Hurricanes in the Intra-Americas Sea. In: Memories of the XI Congreso Latinoamericano e Ibérico de Meteorología y XIV Congreso Mexicano de Meteorología , CD electronic document.

Singh, H. K. A., Hakim, G. J., Tardif, R., Emile-Geay, J., and Noone, D. C.: Insights into Atlantic multidecadal variability using the Last Millennium Reanalysis framework, Clim. Past, 14, 157–174, https://doi.org/10.5194/cp-14-157-2018, 2018.

---

## Author Comment (AC8)

Last comment on: "The Earth's climate lagged, recurrent and non-linear solar and lunar multi-millennial scale responses: An oceanic hypothesis, evidence, verifications and forecasts" (document ESD-2021-84)

by Jorge Sánchez-Sesma

**INDEX**

**A. Answers to referee #1**

In the following the referee #1 comment (RC1), in brown color, is followed with my answers(in red color)

Review of the manuscript '**The Earth's climate lagged, recurrent and non-linear solar and lunar multi-millennial scale responses: An oceanic hypothesis, evidence, verifications and forecasts**' by Jorge Sánchez-Sesma (Independent researcher).

The author presents the 'Lagged & Recurrent Climate Hypothesis' (LRCH), claiming that much of the climate millennial scale variability (with periods between some hundreds of years and a few millennia) is driven by intrinsic solar variability and extreme tides associated with Sun-Earth-Moon orbital coincidences, reoccurring at certain repeated periods. Those extreme tides generate cooling of the sea surface by increased vertical mixing. The lagged response of that forcing in the surface temperature is driven by the Ocean conveyor belt (OCB) with the lag being the 'age of the water' roughly measured by the 'oceanic accumulated distance from the tropical East Pacific'.

The **above mechanistic theory** is quite **simplistic** and even not **falsifiable** according to a fundamental Popper's criterium (https://en.wikipedia.org/wiki/Falsifiability) for the validity of scientific theories, The justification of the theory is based on: 1) ad-hoc, physically not well-grounded arguments, generally taken from **ancient literature** and 2) a set of (nonstatistically validated) fitting regressions using over-simplistic linear modulated/lagged models using as predictors extrapolated data by Fourier analysis of the total solar irradiance (TSI) at every 9.5kyr and of the tidal forcing at every 5kyr. **The 9.5kyr solar recurrence** is misleadingly supported by the same author in Sesma 2016 (SS16 in the manuscript) by a spectral analysis without any statistical significance study analysis of the spectral peaks. That replication (both in the past and future) for a multiple number of Fourier analysis timeseries length (9.5 Kyr) is extremely dangerous due to the high dependence of phases of Fourier components from the analysis period. Moreover, **the concept of 'age of water' is quite ambiguous because the OCB** is a graph with multiple bifurcations, dispersion, stochasticity etc., hence not allowing to estimate the age of water by simple backward cinematics.

1. **Simplistic** is a desirable characteristic of any climate variability explanation.
2. The **Falsiability** of a theory and/or hypothesis will not be required, because my paper will present only "empirical evidences" of climate lagged variations.
3. However the background section of my work is valid, because the supposed "**ancient literature**" could be justified: Although those literature is coming from around 50 or more years ago, at that time the isotopic information began to be available, and the human mind development (based on anthropological knowledge), has been in similar and possible better condition than present times, because they spent more time to analyze the climatic processes. For instance, the Albert Einstein studies developed at the earlier 20[th] century, would be discarded by the Referee #1 as "ancient literature".
4. The recurrences of solar activity at every 9.5kyr and of the tidal forcing at every 5kyr, **will not be considered in final version of the ESD-21-84 document.** However, shorter recurrences, around 900 and 177 years, previously detected in climate and astronomical studies are going to be employed.
5. Although the concept of '**age of water**' was not employed, in the original paper it was proposed an accumulated distance travelled by surface flows of the OCB, from eastern equatorial Pacific, and a lag respect the TSI record. However the new approach in my new paper will not depends objectively from these variables. A new variable will be

analyzed: it is a relative lag/lead of the GT climate records.

The author commits too many methodological and technical flaws in his presentation and discussion of results.

Moreover, the author does not cite or contextualize in his research, related (well supported and methodologically much more advanced) works studying the possible origins of the millennial variability at scales slower than that of Milankovitch forcings. In fact, it is an open discussion, far from being at the end, if the above referred variability comes mostly from the solar/tidal forcing or from the internal slow (not externally forced) variability of the climate system (Soon et al. 2016) or even from magnetic forcing.

**The author explained millennia scale climate variations mainly as a response to astronomical forcing. Through the ocean flows the astro-forcing is moved and delayed in its influences. Although it is a Simplistic** explanation, but also is a desirable explanation of climate varibility that will complement the actual models and, surely, will promote better understanding.

**The author's belief both in the LRCH and his simplistic model, led him to produce forecasts of the global surface temperature for the next decades, in a clear confrontation and contradiction with the decadal forecasts of the much more sophisticated and physically well-grounded models presented worldwide in the IPCC reports.**

**For science development, confrontation and contradiction of ideas is a desirable situation, speciallly when the future of global climate is analyzed. Simplistic models can contribute with more fresh ideas in the present "conundrum" of climate models that was explained in the first version of my paper.**

Despite the author's recognition in the conclusion of the paper, that **his model deserves much more work and physical evaluation, the present work cannot be considered appropriate to be published in the scientific journal ESD.**

**My paper, in its final version, will show a new empirical approach that needs to be taken into account to better appreciate the complexity of natural climate variations. It is not an ended work but, surely, it will open many efforts and contributions to develop, together, a complete and robust climate science theory that provide reliable climate scenarios.**

Next, there are presented some of the major technical and methodological pitfalls of the manuscript:
1) **Figure 1a is totally inappropriate**
2) **Figs 3a and 3b are not legible.** The adapted Fig. 3c is too much stretched and without legibility.
3) **The model in Eq. (1)** of the climate proxy variable, uses the solar proxy variable S(t) which is supposed itself to be self-similar (with a 9.5kyr period). **Therefore, any slower variability with periods larger than 9.5kyr is ignored.** The regression score (e.g. MSE) of the fitting is never shown systematically for the tested climatic proxies neither compared against any 95%-expected score of a null-hypothesis model (e.g. an red noise AR1 model). Therefore, the absence of statistical validation is not allowing any attribution arguments of the variability to the tested external forcing.
4) **Eq. (5)** is completely out of the context and add nothing about the tidal forcing particularities.
5) The details of models (Section 4) should preferentially **be presented in a Table.**

6) Fig.4 is not clear at all. All panels (except panel b) present two curves which are not explained anywhere, even in the caption. Observed proxy data of some of the timeseries does not match perfectly with graphs of the papers from which they where obtained. For instance: a) the Record crbT (green curve of fig 4a) seems to be a degraded representation of data presented in Fig. 2c of Weijers, et al.(2007). b) Record jriT (fig. 4b) representing the Northeast Antarctic Peninsula temperature, jriT (Mulvaney et al., 2012) should represent in effect an anomaly (with respect to the period 1961-1990) in the Antarctic James Ross Island, presented in Fig. 3 (black curve) of Mulvaney et al., 2012. The curves don't match.

7) The **Figs 4d and 5d**, supposedly to present the global temperature anomaly do not match. Difference between curves are not explained anywhere.

8) **Fig. 8a** is an appropriate stretched copy-paste figure without any information of the time scale.

9) **Figs. 8b and 8c** presumably showing the contributions of the solar and tidal influences to the global temperature, are not all well explained. The explained variances are not computed and tested against null hypothesis.

10) The manuscript is full of confusing acronyms, which are not well defined in the text, like NonRad&NonLinear , TidalNonLin , climate(laggedTSI).

11) The description of the **tidal contributions** in the Discussion section 5, pg. 13 lines 23-34 is quite confusing.

12) **Figs 10a,b** present forecasts of the global temperature up to 4000 cal A.D. The author claims that the use of the NASA's solar system astronomical model represents an independent verification of the model. For instance, the time-series issued from models Trend + mTSI + TidalNonLin (blue line) and Trend + mTSI + TidalNASA (cyan line) appear to be quite different and hence not corroborating the empirical proposed model Trend + mTSI + TidalNonLin.

1. **Figures 1a, will be eliminated.**
2. **Figure 3, will be re-made, considering the comment about, but focused on the Atlantic ocean and their "Gulf stream" and other surface currents.**
3. **Equation 1 indicates a model for the lagged and linear reconstructed solar influences on climate (not solar model). Statistical and spectral validations are conditions and test that will be required with more precise models and information.**
4. **Equation 5 indicates a model for astronomical aceleration of Earth masses due to solar and lunar influences. It provides an objective approximation for the original tidal influences.**
5. **Main results will presented in a Table.**
6. **Figures 4 and 5 will be re-made considering only four records (crbT, Antarctic tempertaures, GrT and GT) withouth the jriT record,**
7. **Figures 4 and 5, related with the GT record will be improved and reformatted.**
8. **Figure 8 will be reformatted, eliminating the tidal recurrences, and keeping only the part of the Fig 8c, related with the objective evaluation of "tidal forcing" based on Eq. 5.**
9. **No one statistical test is applied in this work, but confirmations with multiple and independent lines of evidence are developed.**
10. **All the acronyms employed will be reviewed.**
11. **The description of tidal influences (only those from Eq.5) will be reviewed.**
12. **Figure 10, will be re-made, and the corroboration of results, associated with GT forecast will be focused on the 21[st] century. It will be presented in additional figures and will consider more than 10 independent climate records.**

References

Soon, Willie; Velasco Herrera, Victor M.; Selvaraj, Kandasamy; Traversi, Rita; Usoskin, Ilya; Chen, Chen-Tung Arthur; Lou, Jiann-Yuh; Kao, Shuh-Ji; Carter, Robert M.; Pipin, Valery; Severi, Mirko; Becagli, Silvia (2014). A review of Holocene solar-linked climatic variation on centennial to millennial timescales: Physical processes, interpretative frameworks and a new multiple cross-wavelet transform algorithm. Earth-Science Reviews, 134(), 1–15. doi:10.1016/j.earscirev.2014.03.00

**B. Answers to referee #2**

In the following the referee #2 comment (RC2), in brown color, followed with my answers(in red color)

Referee comment on "The Earth's climate lagged, recurrent and non-linear solar and lunar multi-millennial scale responses: An oceanic hypothesis, evidence, verifications and forecasts" by Jorge Sánchez-Sesma, Earth Syst. Dynam. Discuss., https://doi.org/10.5194/esd-2021-84-RC2, 2022

This manuscript seeks to address an extremely relevant scientific problem to the Earth System Dynamics community. Hence, in terms of placement within the scope of the journal, it would be perfect - provided the research would have been conducted in a thorough comprehensive and fail-proof or at least falsifiable manner.

However, notwithstanding the intellectually fertile ideas and insights as expressed in the conjectures and hypothesis laid out for investigation, formally speaking **the study undertaking lacks fundamental scientific grounds to take off as a full-fledged research study. As also already pointed by another reviewer, whose words I fully endorse and hence will not repeat for obvious reasons, there are profound shortcomings and severe hindrances at both technical and scientific levels that make it unfeasible to simply amend in view of a possible publication.**

I have taking into account, in a detailed manner, the comments and suggestions of Reviewer #1,

I am aware of the author's keen efforts to further clarify and improve the manuscript. However, while I sympathise with such efforts and persistence, unfortunately I am so sorry to say that the fatal concerns are not yet sufficiently addressed. **This study needs to go back to the drawing board and reframed from its very foundations, rather than undergoing amendments over what are unstable principles, assumptions and procedures.** Therefore, to that regard, **my recommendation is for the author to take the fertile insights towards producing a clean, sharp, effective study.**

Following these excellent ideas, I have been working in a new version of my paper, taking into account fully the comments and suggestions of Reviewer #2,

The present preprint is citeable and holds the proof of the precedence of the raised ideas and insights. But these need to be thoroughly investigated with technically sound methodologies to provide results that can provide a scientifically sound set of results that can give confidence about the proposed contribution. Until that happens, this study conveys a fertile albeit speculative exercise that is not yet sufficiently close to physical consistence to be deemed appropriate for final publication at Earth System Dynamics.

All in all, **the problem is not on the hypothesis raised by the author, and which should indeed merit further investigation.** It is about how such hypothesis are scientifically worked towards providing a robust contribution to the advancement of knowledge beyond a speculative theoretical exercise grounded on debatable foundations that themselves need to be properly investigated and potentially validated in perhaps a seminal study on its own.

It is important to emphasize that in the new version of my paper, no one hypothesis is made, it is focused in evidences and their verifications, that will help as initial ideas of new approaches in climate research.

**C. General answers to both referees**

The final answers to both reviewers are as follows:

1. I am going to fully modify the original article considering each one of the observations made by both Referees. To do this, I will proceed to the following:
a) I will not mention (neither use) solar and lunar millennia-scale recurrences,
b) I will reestimate the future evolution of the global temperature considering reconstructed and published records of the TSI and the Sun-Moon gravitational accelerations on the terrestrial mass, an amplification is applied during the last centuries supposing volcanic and solar influences (in this document new elements and evidences about volcanic forcing are presented, that empirically justify this amplification) and the remaining residual oscillations will be analyzed through multidecadal-scale analogs. The results will provide empirical scenarios of GT for the 21$^{st}$ century (ES-GT-21$^{st}$C).

2. I will give a new approach to the article:
a) I will present as evidence of the climate delays based on:
- the records in Wiejers et al., 2007, that show increasing relative lags from the crbT record, going from the South to the North Pole, following the Surface flows over the Atlantic ocean.
- the results published by Shakun 2012, which show the relative thermal delay of the Northern Hemisphere with respect to the Southern Hemisphere;
- the new results published by Kaufman et al., 2021, which provide zonal estimations that show the lagged responses of GT respect to South pole records.

3. I will include more climatic records directly related to the Atlantic Ocean and especially in the North Atlantic, to verify the ES-GT-21$^{st}$C for the coming decades. Special contributions will be including: a) rcp0.0 extrapolated IPCC scenarios complemented with constrained scenarios (Risbes et al. (2021); hereafter R21), b) Potential volcanic contributions (Bethke et al. (2017); hereafter B17), c) Antarctic and tropical contributions, and d) tropical, western and northern, North Atlantic records.

4.. Finally I want to:
a) Mention the need, and oportunity, to discuss new options for climate analysis.
b) Thank to referee #2 for his wise comments about making an article clearer, simpler and more direct. Those ideas will be fully considered in the last version

**D. Description of the final document**

D.1 New tittle: *Empirical evidence of astronomical lagged influences on climate: A consequent and multi-verified, global 21st century forecast*.

D.2 New approach: It is focusing around global temperature. In addition to lagged responses to solar and tidal forcing that result in the GT scenario for the 21$^{st}$ century (GT-21$^{st}$ Cent), in this updated document, multi-evidences to support GT-21$^{st}$ Cent, were included.

D.3 New and important evidences. In the following a description of these evidences are provided:

*D.3.1: Evidence of different lagged responses in climate.*
Respect the crbT record, a continental tropical from Congo River Basin (CRB)[W07], the climate variations of Antarctic climate(Dome C), Eastern Equatorial Atlantic SST (S05), Greenland (GISP2) and Global (S12, M13) temperatures show increasing relative lags. It is shown in Figure 1.
It is important to mention here that the W07 and S05 records are coming from the same core of deep sea sediments, lipids indicate the tempearture of SAT in the CRB but the isotopic values from foraminifera skeletons indicate the SST just in front the Congo River discharge. These variables indicate different processes with relative lags of more than 2000 years The crbT indicate the continental quick response to solar forcing, while the crdSST indicate the movement of sea water from Pacific, Indian and Antarctic oceans before to enter the Atlantic ocean.
Respect the SHT(Shakun et al., 2012) and South pole (Kaufman et al., 2020) records, the NHT (Shakun et al., 2012) lags more than 1000 years. It is shown in Figure 2.

*D.3.2: Empirical forecast for the next centuries of GT (lagging South Pole and Caribbean climates).*
Taking into account three south Pole climate records an empirical GT forecast is developed. We use the EDML record, the SST west of Antarctic Peninsula (Shevenellet al., 2012), and the zonal integration (60-90 °S) of proxy information (Kaufman et al., 2021). A Cariaco SST record is also analyzed (Black et al., 2004). These empirical modellings are shown in Figure 3, and their details are shown in Figure 4.

*D.3.3: Empirical modelling of GT (leading Hydroclimate)*
In contrast Titanium and iron concentration data from the anoxic Cariaco Basin, off the Venezuelan coast, that can be used to infer variations in the hydrological cycle over northern South America during the past 14,000 years with subdecadal resolution (Huag et al., 2001), show different timing respect GT. These two independent records for Titanium and Iron content of deep sediments over the Holocene, indicate that GT leads them (for more than 200 years). Figures 5 and 6 ilustrate the connection between GT and those minerals contents.

The GT lags respect to South pole records indicate the ocean currents that takes heat and mass in a time journey of more than a millennium. In contrast, the GT leads respect to the Cariaco mineral records indicate the ocean-atmosphere currents from northern latitudes toward the tropics in a journey that includes hydrologic response that takes time of more than two centuries.

*D.3.4: Empirical verification of previous GT forecast for the next decades.*
Taking into account several independent climate records, their climate influences were estimated applying a linear transformation, a linear trend and a constant lag.

D.3.4.1 Estimations of GT based on South pole and Cariaco
The South pole records indicate GT lags (for more than 1000 years) with consequent forecast of GT with a cooling for the next centuries. The Caribean record indicate lower GT lags (for more than 600 years) with consequent forecast of GT with a cooling, at least for the next centuries. Figure 7 and 8 show these independent two GT forecasts.

D.3.4.2 Estimations of GT based on other Caribbean, North Atlantic and Arctic records
Figure 9 display 6 different forecast and its statistical range of values (mean, mean+StdDev, mean-StdDev) for the next decades. These variables have been discussed and analyzed in previous comments.

D.3.4.3 Updated IPCC 2013 TAS extrapolated scenarios to rcp0.0
Considering the GHG hypothesis is not able to apply in a climate system with the detected lagged climate responses, a simple linear extrapolations of the rcp0.0 was evaluated in a previous comment with IPCC 2013 unconstrained values. It should be noted that this extrapolation is valid at the end of the 21$^{st}$ century.
        However, here we add another evaluation considering the constrained values (Risbes et al. (2021); hereafter R21). R21 have pointed out: "Many studies have sought to constrain climate projections based on recent observations. Until recently, these constraints had limited impact, and projected warming ranges were driven primarily by model outputs....we use the newest climate model ensemble, improved observations, and a new statistical method to narrow uncertainty on estimates of past and future human-induced warming..... Our results suggest that using an unconstrained multimodel ensemble is no longer the best choice for global mean temperature projections..."
        We analyzed the rcp0.0 constrained scenarios for AR5 during the last two decades of the 21$^{st}$ century (2080-2100). The results confirm the rcp0.0 unconstrained scenarios for the AR5 evaluated previously. Both constrained (R21), with range and mean values, and unconstraind rcp0.0 (IPCC2013) scenarios with ensemble values, show compatible behaviour. The comparison is shown in Figure 10.

D.3.4.4 Extratropical Northern Hemisphere Temperature (ExTropNHT) extrapolated values for the next centuries.
Based on a detected recurrence of climate values with period of around 900 years (Schulz and Paul, 2001) and a possible orbital forcing independently estimated (Loutre et al., ) the extrapolation of the ExTropNHT reconstructed values (Buentgen et al, 20XX) was developed (See Appendix A). Its lagged influences of ExTropNHT on GT, around 20 years later, provides an additional verification for GT21st.

D.3.4.5 Influences of Volcanic Activity in the next decades.
Based on a detected recurrence of climate values with period of around 177 years the extrapolation of the N3v reconstructed values (Mann et al, 2005) was developed (See Appendix B). Its recurrent influences of N3v on GT, provides an additional verification for GT21st.

**E. Last remarks**

We have shown evidences of
1. Lagged climate processes in different scales: multimillenial, multicentennial and multidecadal.
2. Natural recurrences of climate processes with periods around 900 and 177 yrs.
3. Two regional studies of ocean deep water sediments, at Congo river discharge, and at Cariaco basin, provide information in different media (lipids, isotopic and mineral content) to different climate processes with different relative lags, of around millennia and millenium, respectively, that show parallel processes in climate with different timing.
4. Multiple climate processes that happen before the GT provide information not only to forecast a gentle cooling for the rest of the 21$^{st}$ century, but also, to forecast a similar gentle cooling and oscilations for the next 1 and a half millennium.
5. The importance of volcanic forcing and its recurrent responses, with periods of around 177 yrs, has been shown in the ENSO-N3v simulated records that clearly influence GT over the last century, not only provide elements for a special consideration (magnification) of the GT responses over the last centuries (made in the GT empirical modelling in the first document), but also provides values of volcanic influences for the rest of the 21$^{st}$ century.
6. The volcanic empirical modelling of climate responses, are similar in magnitude and timing to those analyzed by Bethke et al. (2017: hereafter B17) and published in Nature Climate Change, however the sign is different. While the B17 publication indicates negative contributions of volcanic activity, the empirical modelling generates in a simplistic form, but sustained with one of the best (simplified) models of ENSO, postive contributions of the volcanic activity. The key point of these differences is the "Ocean heat thermostat" (OHT) named by Clement et al. (1996) that discovered that the mean surface temperature change result of the ocean-atmosphere system at the tropics shows cooling (warming) when the radiation increase (decrease).

Taking into account the original paper ESD-2021-84 and all the comments developed, but specially this last one comment, I would like to emphasize that the natural processes of the Earth's climate system requieres this , and similar documents, to be better understood, and then better modelled.

Many empirical lines of evidences have shown that the rest of the 21$^{st}$ century will be characterized by a cooling processes. The IPCC warming for the same period present great modelling limitations, the mentioned "holocene conundrum" and the modelling of volcanic impacts on GT for the 21$^{st}$ century have shown important limitations. However, the most important limitation is that the ocean surface and at middle depths currents, have not being well considered as we have shown in all of our documents around ESD-2021-84.

**F. References**

Bethke, I., Outten, S., Otterå, O. et al. Potential volcanic impacts on future climate variability. Nature Clim Change 7, 799–805, https://doi.org/10.1038/nclimate3394, 2017.

Black, DE, Robert C. Thunell, Alexey Kaplan, Larry C. Peterson, and Eric J. Tappa. 2004. A 2000-year record of caribbean and tropical north atlantic hydrographic variability, Paleoceanography, 19, PA2022, doi:10.1029/2003PA000982, 2004

Büntgen U, Arseneault D, Boucher É, Churakova (Sidorova) OV, Gennaretti F, Crivellaro A, Hughes MK, Kirdyanov A, Klippel L, Krusic PJ, Linderholm HW, Ljungqvist FC, Ludescher J, McCormick M, Myglan VS, Nicolussi K, Piermattei A, Oppenheimer C, Reinig F, Sigl M, Vaganov EA, Esper J (2020) Prominent role of volcanism in Common Era climate variability and human history. Dendrochronologia 64: 125757

Donnelly, J., Woodruff, J. Intense hurricane activity over the past 5,000 years controlled by El Niño and the West African monsoon. Nature 447, 465–468, https://doi.org/10.1038/nature05834, 2007.

Donnelly, J.P., A.D. Hawkes, P. Lane, D. MacDonald, B.N. Shuman, M.R. Toomey, P.J. van Hengstum: Climate Forcing of Unprecedented Intense-Hurricane Activity in the Last 2,000 Years: Earth's Future (AGU) 3, doi:10.1002/2014EF000274, 2015.

Fairbridge, R. W., and J. E. Sanders, The Sun's orbit, A.D 750–2050: Basis for new perspectives on planetary dynamics and Earth-Moon linkage, in Climate—History, periodicity, and predictability, edited by M. R. Rampino et al., pp. 446– 471, Van Nostrand Reinhold, New York, 1987.

Haase-Schramm, A., F. Böhm, A. Eisenhauer, D. Garbe-Schönberg,W.-C. Dullo, and J. Reitner: Annual to interannual temperature variability in the Caribbean during the Maunder sunspot minimum, Paleoceanography, 20, PA4015, doi:10.1029/2005PA001137, 2005

Haug G.H., Konrad A. Hughen, Daniel M. Sigman, Larry C. Peterson and Ursula Röhl:  Southward Migration of the Intertropical Convergence Zone Through the Holocene, Science, Vol 293 (5533) 1304-1308, doi:10.1126/science.1059725, 2001

IPCC-AR5 2013 RCP

Kaufman, D., McKay, N., Routson, C., Erb, M., Davis, B., Heiri, O., Jaccard, S., Tierney, J., Dätwyler, C., Axford, Y., Brussel, T., Cartapanis, O., Chase, B., Dawson, A., de Vernal, A., Engels, S., Jonkers, L., Marsicek, J., Moffa-Sánchez, P., Morrill, C., Orsi, A., Rehfeld, K., Saunders, K., Sommer, P., Thomas, E., Tonello, M., Tóth, M., Vachula, R., Andreev, A., Bertrand, S., Biskaborn, B., Bringué, M., Brooks, S., Caniupán, M., Chevalier, M., Cwynar, L., Emile-Geay, J., Fegyveresi, J., Feurdean, A., Finsinger, W., Fortin, M., Foster, L., Fox, M., Gajewski, K., Grosjean, M., Hausmann, S., Heinrichs, M., Holmes, N., Ilyashuk, B., Ilyashuk, E., Juggins, S., Khider, D., Koinig, K., Langdon, P., Larocque-Tobler, I., Li, J., Lotter, A., Luoto, T., Mackay, A., Magyari, E., Malevich, S., Mark, B., Massaferro, J., Montade, V., Nazarova, L., Novenko, E., Pařil, P., Pearson, E., Peros, M., Pienitz, R., Płóciennik, M., Porinchu, D., Potito, A., Rees, A., Reinemann, S., Roberts, S., Rolland, N., Salonen, S., Self, A., Seppä, H., Shala, S., St-Jacques, J., Stenni, B., Syrykh, L., Tarrats, P., Taylor, K., van den Bos, V., Velle, G., Wahl, E., Walker, I., Wilmshurst, J., Zhang, E., Zhilich, S., 2020. A global database of Holocene paleo-temperature. Scientific Data 7, 115. doi: 10.1038/s41597-020-0445-3

Kaufman, D., McKay, N., Routson, C., Erb, M., Dätwyler, C., Sommer, P., Heiri, O., Davis, B., 2020. Holocene global surface temperature: A multi-method reconstruction approach. Scientific Data 7, 201. doi: 10.1038/s41597-020-0530-7

Keigwin L.D., The Little Ice Age and Medieval Warm Period in the Sargasso Sea, Science, 274 (5292) 1504-

1508. 1996.

Kobashi, T., Kawamura, K., Severinghaus, J. P., Barnola, J-M, Nakaegawa, T., Vinther, B.M., Johnsen, S.J., Box, J.E., High variability of Greenland surface temperature over the past 4000 years estimated from trapped air in an ice core, GRL, doi:10.1029/2011GL049444, 2011.

Kobashi, T., Goto-Azuma, K., Box, J. E., Gao, C.-C., and Nakaegawa, T.: Causes of Greenland temperature variability over the past 4000 yr: implications for northern hemispheric temperature changes, Clim. Past, 9, 2299–2317, https://doi.org/10.5194/cp-9-2299-2013, 2013.

Kobashi, T., Menviel, L., Jeltsch-Thömmes, A., Vinther, B. M., Box, J. E., Muscheler, R., Nakaegawa, T., Pfister, P. L., Döring, M., Leuenberger, M., Wanner, H., Ohmura, A.: Volcanic influence on centennial to millennial Holocene Greenland temperature change, Sci Rep, doi:10.1038/s41598-017-01451-7, 2017.

Loutre, MF, A Berger, P Bretagnon & P-L Blanca: Astronomical frequencies for climate research at the decadal to century time scale, Climate Dynamics volume 7, pages181–194. 1992.

Mann, M.E., M.A. Cane, S.E. Zebiak, and A. Clement: Volcanic and solar forcing of the tropical Pacific over the past 1000 years. J. Climate, 18, 447–456, 2005.

Rayner N. A., D. E. Parker, E. B. Horton, C. K. Folland, L. V. Alexander, D. P. Rowell, E. C. Kent, A. Kaplan, Global analyses of sea surface temperature, sea ice, and night marine air temperature since the late nineteenth century, J. Geophys. Res., 108 (D14), 4407, doi:10.1029/2002JD002670, 2003.

Ribes A, Qasmi S, Gillett NP. Making climate projections conditional on historical observations. Sci. Adv.;7:eabc0671. doi: 10.1126/sciadv.abc0671, 2021.

Russell, J. M., and T. C. Johnson: Late Holocene climate change in the North Atlantic and equatorial Africa: Millennial-scale ITCZ migration, Geophys. Res. Lett., 32, L17705, doi:10.1029/2005GL023295, 2005.

Schefuß, E., Schouten, S. & Schneider, R. Climatic controls on central African hydrology during the past 20,000 years. Nature 437, 1003–1006. https://doi.org/10.1038/nature03945, 2005.

Schulz, M., Paul, A. (2002): Holocene climate variability on centennial-to-millennial time scales: 1 Climate records from the North-Atlantic realm. In: Wefer, G. et al. (eds.): Climate development and history of the North Atlantic realm. Springer, Berlin, pp.41-54.

Shevenell, A. E., Ingalls A. E., Domack E. W. & Kelly C., Holocene Southern Ocean surface temperature variability west of the Antarctic Peninsula, Nature, 470, doi:10.1038/nature09751, 2011.

Weijers, Johan W H; Schefuß, Enno; Schouten, Stefan; Sinninghe Damsté, Jaap S: Coupled Thermal and Hydrological Evolution of Tropical Africa over the Last Deglaciation. Science, 315(5819), 1701-1704, https://doi.org/10.1126/science.1138131, 2007.

**G. Figures**

[Figure]

**Figure 1. Five climate records, four (A, B, C and D) adapted from Wiejers et al., 2007 and one (E), the global temperature (GT) from Shakun et al, 2012 and Marcott et al, 2013. Two are coming from GISP2 and EPICA Dome C ice cores from polar regions (in gray color), and two are coming from the same sediment core from tropical regions (Congo river basin Temperatures [crbT] in red color, and Atlantic Eastern Tropical SST just in front the Congo river dischage [crdSST], in blue color). The GT record (E) appears to the last climate record, its variations are almost in sinchrony (some decades of difference) with GISP2. These records EPICA, SST, GISP2 and GT are showing relative lags respect the crbT of around 2400, 3000, 4000 and 4000 years, respectively.**

[Figure]

a)

[Figure]

b)

[Figure]

c)

**Figure 2.  Comparison of NHT and adjusted SHT with linear transformations and lags of around 1400 and 1380 years for the a) Shakun et al. 2012, and b) Kaufman et al., 2021,  reconstructions, respectively; c) a zoom of b) indicated decreasing temperatures for the early and middle of the 21st century (-50 to -150 years BP).**

[Figure]

**d)**

**Figure 3. Comparison of GT updated record (see Appendix A of the first version of thiswork) with four adjusted and lagged climate records of temperature coming from South pole and Caribbean. a) EDML record, b) SST west ofAntarcticPeninsula (Shevenellet al., 2012), and c) zonal integration (60-90 °S) of proxy information ( Kaufman et al., 2021), and d) Cariaco SST (Black et al., 2004). Note that: 0 KyBP=1950AD**

[Figure]

**Figure 4. A zoom of forecast based on South Pole climate (Figures 3) covering the period 1-(-0.5) Kyr BP [950-2450 AD].**

[Figure]

a)

[Figure]

b)

**Figure 5. Global Temperature reconstructed compared the Cariaco a) Ti% and b)Fe content reconstructed from deep sea-sediments (Huag et al., 2001). The GT record leads both Cariaco mineral content by more then 200 years.**

[Figure]

a)

[Figure]

b)

**Figure 6. A zoom of Figures 5 covering the last 2000 yr. a) Ti , b) Fe.**

[Figure]

a)

[Figure]

b)

[Figure]

c)

**Figure 7. Integration of global temperature Forecasts based on South Pole records. a) Integration of SP records (Figures 3 a, b and c and 4 a, b and c), b) Mean values for the records shown in a), and c) a zoom 1850-2100 of b)**

[Figure]

a)

b)

**Figure 8. Global temperature Forecasts based on Cariaco SST record. a) GT forecast baed on Cariaco SST (Black, et al., 2004), and c) a zoom 1850-2100 of b)**

[Figure]

a)

b)

c)

**Figure 9. Different GT forecasts for the 21st century. a) GT climate scenarios based on tropical and Northern Atlantic climate records, b) their mean and standard deviations range and c) their comparison with previous GT scenario (ESD-2021-84)**

[Figure]

**Figure 10. Different GT forecasts for the 21$^{st}$ century. a) GT climate scenarios extrapolated to rcp0.0 and complemented with constrained scenarios, b) previous GT scenario (ESD-2021-84) and complemented with multiple records (see Fig. 9)**

[Figure]

a)

b)

**Figure 11. Different GT forecasts for the 21ˢᵗ century. a) GT climate scenarios based on an Antarctic record, b) previous GT scenario (ESD-2021-84) and complemented with multiple records (see Fig. 9)**

[Figure]

a)

b)

**Figure 12. Different GT forecasts for the 21st century. a) GT climate scenarios based on extratropical Northern Hemisphere reconstructed record, b) their comparison with previous GT scenario (ESD-2021-84), rcp0.0 previously presented and complemented in this work with constrained scenarios.**

[Figure]

**Figure 13. Different GT forecasts for the 21st century. a) GT climate scenarios based on PuertoRico extreme tropical cyclones (PR-xtrTC) reconstructed record (Donnelly et al., 2007), b) their comparison with previous GT scenario (ESD-2021-84), rcp0.0 previously presented and complemented in this work with constrained scenarios.**

[Figure]

a)

b)

**Figure 14. Different GT forecasts for the 21st century. a) GT climate scenarios based on Cariaco SST reconstructed record (Black et al., 2004), b) their comparison with previous GT scenario (ESD-2021-84), rcp0.0 previously presented and complemented in this work with constrained scenarios.**

[Figure]

**Figure 15. Different GT forecasts for the 21st century. a) GT climate scenarios based on Montego-Bay Jamaica SST reconstructed record (Haase-Schramm et al., 2005), b) their comparison with previous GT scenario (ESD-2021-84), rcp0.0 previously presented and complemented in this work with constrained scenarios.**

[Figure]

[Figure]

**Figure 16. Comparison of a) GT forecast 2020-2100 made in ESD-2021-84 and this last comment, with b) a model of GT based on ENSO volcanic response (SST N3v) simulated with the ZC model by Mann et al. (2005). For additional comparison volcanic influences on climate (considering model shown in b) and those from Bethke et al., (2017) see Appendix B.**

**H. Appendices**

**APPENDIX A:** Global Temperature forecast based on the extrapolation of Northern Hemisphere Extratropics Temperature (NHExtrT) [Büntgen et al, 2020]

The recent NHExtrT climate reconstruction for the last 2000 years (Büntgen, et al., 2021) is analyzed and extrapolated here. These authors, have reconstructed the full range of past Northern extra-tropical summer temperature variation from 1 to 2010 CE. They apply a novel ensemble climate reconstruction approach on updated tree-ring width (TRW) chronologies from high-elevation/-latitude sites across the Northern Hemisphere. In order to better visualize recurrent pattern the record and its moving standar deviation, each 20 years, are shown in Figure A1.

In the same Figure A1, the NHxtrT and its StdDev records are modelled with analogs lagged around 900 years. The Figure A1a shows the NHxtrT and its analog model (the same record lagged 917 years). The Figure A1b shows the StdDev(NHxtrT) and its analog model (the same record lagged 900 years).

[Figure]

a)

[Figure]

b)

**Figure A1. NHxtrT a) values and b) stddev records and their analog models.**

The lags of around 900 years in both shown models of NHxtrT, could be justified by detected recurences both in climate records (Shulz and Paul, 2002) and in orbital high-frequency variability analysis (Loutre et al. 1992).

Based in the extrapolated NHxtrT a GT forecast is made. It is developed linerly adjusting and lagging the  NHxtrT record to the GT record.

[Figure]

**Figure A2. Global temperature forecast based the linearly adjusted and lagged extrapolated NHxtrT.**

**APPENDIX B:** Global Temperature forecast based on the SST-Niño3 reconstructed with volcanic forcing (Mann et al., 2005)

We analyze an ENSO simulation with solar and volcanic forcing ( Mann et al., 2005). The limited area Cane-Zebiak model provides SST evolution over the tropical Ocean-atmosphere model forced by solar and volcanic activity. This only-volcanic-forcing modelling emphasize the intermittent and aparently errative volcanic activity and results in ENSO variation over the last millennium with oscillations of around 177 years. In order to verify the existence of this oscillations an analog model (a linear transformation of the original record) lagged 177 years, it is overlappped with a very good match (See Figure B1). This model is extrapolated to the next 177 years. The 177 year oscillation, and the corresponding lag, can be justified considering the existence a similar recurrence between several of the major planets of the solar system. An averaging of astronomical information (Fairbridge and Sanders, 1987) for multiple Saturno Jupiter Lap (SJL=19.857 yr), 9*SJL=178.713 yr, Neptune Uranus Lap (NUL=171.39 yr), and multiple Uranus Saturn Lap(USL=45.387yr), 4*USL =181.548yr, results in 177.21 yrs.

Figure B2 show the adjusted extrapolated analog model of N3v to GT. The increasing amplitude of the N3v present a very good match with the GT warming of the 20[th] century.

[Figure]

**Figure A2. ENSO Niño3-SST reconstructed response to volcanic forcing (Mann et al., 2005) and its analog model (with an adusted linear transformation and a lag of 177yr).**

[Figure]

**Figure B2. Global temperature forecast based the linearly adjusted N3v analog model.**

The decreasing amplitude of N3v for the next decades, represent not only another empirical forecast of the GT-21st-century, but the negative of the IPCC scenario for volcanic activity (Bethke et al, 2017) leaded around 10 yr. It is shown in Figure B3.

[Figure]

a)

[Figure]

b)

**Figure B3. 21st Century comparison of Climate Volcanic scenarios. a) negative GT detrended model based on N3v (Seee Fig. B2) and b) the estimated Global volcanic scenario (Bethke et al., 2017) (leaded around 10 yr) that is added to each one of the different RCP scenarios.**

**APPENDIX C:** Global Temperature forecast based on the Puerto Rican extreme tropical cyclone reconstructed record (PR-extr-TC) [Donnelly et al., 2015]

The ePR-extr-TC climate reconstruction for the last 5000 years (Donnelly et al., 2007) has been reanalyzed. Donnelly et al., 2015, have updated dates of extreme TC hits in Estearn PR. These sediment size values are depicted in Figure B1.

[Figure]

**Figure C1. PR-xtr-TC  values with updated dates (Donnelly etal., 2015).**

[Figure]

Figure C2. Global temperature forecast based on the linearly adjusted and lagged PR-xtr-TC. **The values for the model are coming from Donnelly et al. (2007 and 2016) and are lagged 110 years.**

**APPENDIX D:** Global Temperature forecast based on the SST of Cariaco Basin (SST-Car-d18O-G.Bull]) Black et al., (2004)

The recent SST-Car-d18OGBull climate reconstruction for the last 2000 years (Black et al., 2004) is analyzed here.

These authors, have reconstructed near-annually resolved oxygen isotope records from planktic foraminifera from the Cariaco Basin that reflect sea surface temperature (SST) and Intertropical Convergence Zone (ITCZ) precipitation-related salinity variations over the Caribbean and tropical North Atlantic spanning the last 2000 years. These reconstructed values are shown in Figure C1.

[Figure]

**Figure D1. SST(d18OG.Bull) values from Black et al. (2004).**

The Cariaco reconstructed record, lagged 670 years and linearly adjusted represent and forecast GT over the next seven centuries. It is depicted in Figure C2.

[Figure]

**Figure D2. GT and its model based on Cariaco SST(SST-Car-d18OG.Bull). The values for the model are coming from Black et al. (2004) and are lagged 670 years.**

**APPENDIX E:** Global Temperature forecast based on the SST of Jamaica (Montego Bay) (SST[Sr/Ca Sclerosponges]) Haase-Schramm et al., (2005)

[Figure]

**Figure E1. Jamaica SST(Sr/Ca) values from Haase-Schramm et al. (2005).**

[Figure]

**Figure E2. GT and its model based on Jamaica SST. The values for the model are coming from Haase-Schramm et al. (2005) and are lagged 65 years.**